# Fluorescent protein lifetimes report densities and phases of nuclear condensates during embryonic stem-cell differentiation

Khalil Joron[1,8], Juliane Oliveira Viegas[2,8], Liam Haas-Neill[3,4,5], Sariel Bier[1,2], Paz Drori[1], Shani Dvir[1], Patrick Siang Lin Lim [2], Sarah Rauscher [3,4,5], Eran Meshorer [2,6,9] ✉ & Eitan Lerner [1,7,9] ✉

Fluorescent proteins (FP) are frequently used for studying proteins inside cells. In advanced fluorescence microscopy, FPs can report on additional intracellular variables. One variable is the local density near FPs, which can be useful in studying densities within cellular bio-condensates. Here, we show that a reduction in fluorescence lifetimes of common monomeric FPs reports increased levels of local densities. We demonstrate the use of this fluorescence-based variable to report the distribution of local densities within heterochromatin protein 1α (HP1α) in mouse embryonic stem cells (ESCs), before and after early differentiation. We find that local densities within HP1α condensates in pluripotent ESCs are heterogeneous and cannot be explained by a single liquid phase. Early differentiation, however, induces a change towards a more homogeneous distribution of local densities, which can be explained as a liquid-like phase. In conclusion, we provide a fluorescence-based method to report increased local densities and apply it to distinguish between homogeneous and heterogeneous local densities within bio-condensates.

The typical intracellular crowdedness is estimated at 20-30% fractional volume occupancy (FVO)[1–4]. However, membrane-less bio-condensates that might have undergone phase separation (PS) may exhibit higher densities and hence higher crowdedness levels. For example, the genetic material in living eukaryotic cells is packaged in the form of chromatin, where compaction and expansion of chromatin structure may regulate gene expression and dictate cellular function. In embryonic stem cells (ESCs) and early embryos, chromatin was shown to have a unique open configuration and a hyper-

dynamic association with chromatin proteins, including histone proteins and heterochromatin proteins (HP), such as HP1[5–7], a phenomenon also observed in cancer stem cells[8]. The unique conformation of chromatin in pluripotent cells is hypothesized to support pluripotency and cell lineage specifications[9]. In recent years, a growing number of studies emerged, which highlight chromatin as a PS nuclear condensate[10], with liquid-like properties at the nanoscale[11], and solid-like properties at the mesoscale[12]. Studies in yeast and fly demonstrated that liquid-liquid PS (LLPS) of HP1 drives

[1]Department of Biological Chemistry, The Alexander Silberman Institute of Life Sciences, Faculty of Mathematics & Science, The Edmond J. Safra Campus, The Hebrew University of Jerusalem, Jerusalem 9190401, Israel. [2]Department of Genetics, The Alexander Silberman Institute of Life Sciences, The Hebrew University of Jerusalem, Edmond J. Safra Campus, Givat Ram, Jerusalem 91904, Israel. [3]Department of Chemical and Physical Sciences, University of Toronto Mississauga, Mississauga, ON L5L 1C6, Canada. [4]Department of Physics, University of Toronto, Toronto, ON M5S 1A7, Canada. [5]Department of Chemistry, University of Toronto, 80 St. George Street, Toronto, ON M5S 3H6, Canada. [6]Edmond and Lily Center for Brain Sciences (ELSC), The Hebrew University of Jerusalem, Jerusalem 91904, Israel. [7]The Center for Nanoscience and Nanotechnology, The Hebrew University of Jerusalem, Jerusalem 9190401, Israel. [8]These authors contributed equally: Khalil Joron, Juliane Oliveira Viegas. [9]These authors jointly supervised this work: Eran Meshorer, Eitan Lerner. ✉e-mail: eran.meshorer@mail.huji.ac.il; eitan.lerner@mail.huji.ac.il

the formation of heterochromatin nano-domains[13,14]. HP1α is commonly associated with silenced heterochromatin regions[15–17], is highly dynamic[18–21], and is considered a crucial factor in chromatin organization and compaction[22–24]. Moreover, HP1α can form LLPS droplets in different cells and species[13,14,22–24]. In mammalian cells, however, HP1α has recently been shown to favor forming condensates via interactions with spatially clustered binding sites (ICBS) without undergoing LLPS[25,26]. Nevertheless, since these studies were performed on differentiated cells, the question remains as to whether HP1 favors forming LLPS- or ICBS-based condensates in undifferentiated cells, especially in pluripotent stem cells, where mesoscale histone dynamics are exacerbated[27]. Recent results show that in undifferentiated ESCs heterochromatin condensates, involving chromocenters, major satellite repeat (MSR) transcripts, and HP1α may be considered as PS condensates with some features of a liquid phase and others that deviate from a liquid phase[28]. Yet, the characterization of the biophysical process of HP1α condensate formation in ESCs before and during differentiation is still incomplete.

A general feature of liquid droplets exhibiting a single phase and separated from bulk liquid, much like an oil droplet in water, is that they should exhibit homogeneous density throughout the interior of the condensate, except, perhaps, for its edges[29–31]. Fluorescence-based reporters of local density or crowdedness within an imaged cell[32,33] could help probe this feature of condensates.

In general, fluorescent proteins (FPs) are commonly used in microscopy for studying protein expression, localization, dynamics, interactions, regulation, and many other biological phenomena by genetic fusion to a protein. From a structural perspective, FPs include an organic fluorophore formed from a triad of residues that react. The fluorophore maturation process occurs inside a β-barrel protein fold, between two α-helices crossing the β-barrel from above and below, and an elaborate network of interactions in the interior of the β-barrel constricts the fluorophore in a well-defined position with minimal degrees of freedom[32–36]. These features render fluorescence intensities from well-folded monomeric FPs as objective reporters of concentration of the FP-fused protein.

Fluorescence is a result of a radiative de-excitation pathway, involving the emission of a photon, which competes with a non-radiative de-excitation pathway, which is influenced by micro-environmental changes in the vicinity of the fluorophore group, among other factors (e.g., temperature, pH). In FPs, such factors can be introduced by an increase in the degrees of freedom of the fluorophore within the β-barrel fold. The overall de-excitation rate is commonly measured via its reciprocal value, the fluorescence lifetime, which is the main observable in fluorescence lifetime imaging (FLIM). Therefore, if elevated levels of local densities or crowdedness lead to an increase in the degrees of freedom in the vicinity of the fluorophore inside the β-barrel fold, which enhance the nonradiative de-excitation rate, then the fluorescence lifetimes of the FPs can be used in detecting these elevated levels of local densities or crowdedness levels at different imaged regions.

Here, we find that significant reductions in the fluorescence lifetime of some monomeric FPs, relative to their typical values, report on densities and crowdedness levels at well-defined FVO values, and provide a working mechanism based on results from experiments and simulations. We use this methodological development and employ FLIM to study and compare the PS behavior of FP-tagged HP1α in mouse ESCs before and after early differentiation. We demonstrate that HP1α condensates in ESCs exhibit density heterogeneity that could not be explained by LLPS into a single-phase liquid-like condensate. However, HP1α condensates in retinoic acid (RA)-treated ESCs or in ESCs in which major satellite repeats (MSR) were depleted, exhibit lower and less heterogeneous densities, suggesting a shift towards a single liquid-like phase.

## Results

### Fluorescence lifetime reductions in FPs due to elevated crowdedness levels

To assess the use of fluorescence lifetimes from monomeric FPs for sensing specific physicochemical conditions, we conducted an array of in vitro characterizations of such recombinant FPs. We measured mCherry fluorescence decays at different concentrations of: (1) small salt molecules (i.e., NaCl), (2) small viscogen molecules, inducing viscosity yet not crowding (i.e., glycerol), (3) small molecules inducing both viscosity and molecular crowding, acting as protein kosmotropes (i.e., trehalose[37]), and (4) macromolecules inducing both viscosity and macromolecular crowding (i.e., macromolecular crowding agents, such as polyethylene glycol, PEG) of different sizes (e.g., PEG 3,350, PEG 6,000, PEG 10,000[38–40]; Supplementary Fig. 1). We fitted the results to a bi-exponential decay model convolved with the experimental impulse response function (Supplementary Eq. (1); Supplementary Fig. 2). The best-fit fluorescence lifetime components and their amplitudes were then used for calculating the mean fluorescence lifetimes (Supplementary Eq. (2)). The results show that different concentrations of NaCl (Fig. 1a) or glycerol (Fig. 1b) do not introduce significant changes or monotonic trends in the values of mean fluorescence lifetimes of mCherry (Fig. 1; Supplementary Table 1). Importantly, the mean fluorescence lifetimes of mCherry in these conditions are around its standard value, 1.4–1.6 ns[41,42]. Using these non-monotonic trends of mCherry lifetimes, we set a lower boundary of 1.44 ns (Fig. 1, dashed red line) as the lowest value for the fluorescence lifetime of mCherry. We then tested for conditions exhibiting fluorescence lifetimes lower than this value. When crowding was introduced, either by the small kosmotrope, trehalose (Fig. 1c), or by the macromolecular crowding agents, PEG of different sizes (Fig. 1d–f), fluorescence lifetimes of mCherry did not decrease from its standard value range at low concentrations. However, at higher kosmotrope and crowding agent concentrations, the results exhibited a monotonic decrease in fluorescence lifetime as a function of concentration (Fig. 1c, f). Converting PEG concentrations to FVO values shows that the monotonic trend in the reduction of mCherry fluorescence lifetimes starts at FVOs of ~30%. To ensure that the observed effect is solely due to changes in fluorescence lifetime, we recorded the steady-state fluorescence spectra of mCherry in these conditions, and the results show that mCherry spectra do not exhibit non-negligible spectral chromatic shifts as a function of the different conditions tested (Supplementary Fig. 3).

Importantly, increasing the concentration of kosmotropes and macromolecular crowding agents also increase the refractive index in the bulk solution. If the refractive index in the vicinity of the fluorophore inside the β-barrel also increases, that could also introduce reduction in the fluorescence lifetime, which can be explained by the Strickler-Berg relationship[30]. Interestingly, this effect did not clearly show up in fluorescence lifetimes of mCherry (Fig. 1b-f), even though the bulk refractive index increases at the tested conditions. We note that this can happen if the local refractive index inside mCherry, in the vicinity to its fluorophore, is not influenced from the bulk refractive index.

Encouraged by the results of mCherry, we performed similar analyses for other common monomeric FPs (e.g., mRFP, mCitrine, eGFP), which exhibited similar trends as mCherry, relative to their standard fluorescence lifetime values (Supplementary Figs. 4–6), albeit with different value ranges. Interestingly, eGFP and mCitrine reported fluorescence lifetime reductions starting from values of ~2.6-2.7 ns[41,43,44] and ~3.4-3.6 ns[45–48], respectively, also as a function of viscosity (Supplementary Figs. 5–6), unlike in mCherry and mRFP. Yet, reductions of fluorescence lifetimes of eGFP and mCitrine below the lifetime values at 50% glycerol still occur in the presence of PEG at crowding levels >30% FVO (Supplementary Figs. 5–6). Could this be due to changes in the local refractive index in the vicinity of the fluorophore

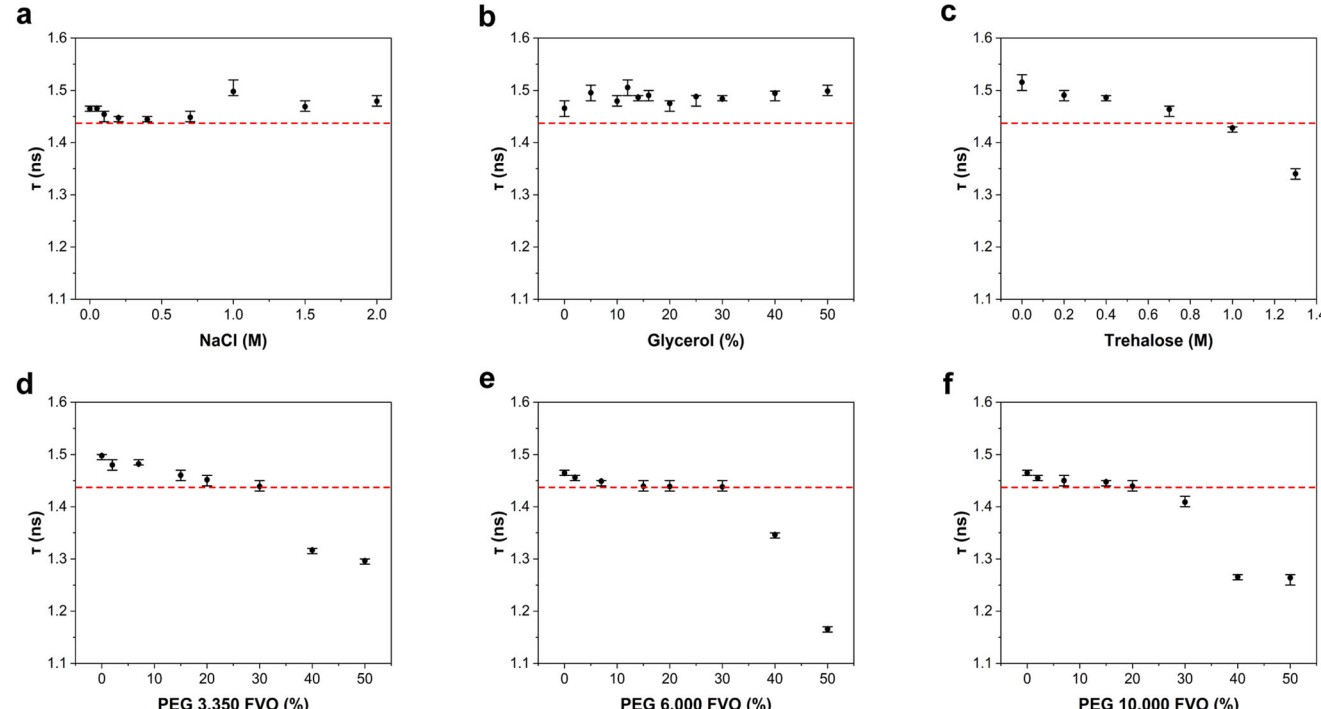

**Fig. 1 | The mean fluorescence lifetime of mCherry at different physico-chemical conditions. a–f** mean fluorescence lifetime (τ) of 100 nM mCherry as a measure of increasing NaCl concentrations **a** Glycerol, **b** Trehalose, **c**, and **d–f** polyethylene glycol (PEG). Crowding exclusively induces lifetime reduction, and only above 30% fractional volume occupancy (FVO). The values and error estimates are based on calculations (Eq. S2) using best-fit values of the recorded fluorescence decays (Supplementary Fig. 1) to a sum of two exponents function (Eq. S1). The values of the intrinsic mean fluorescence lifetimes are reported in Table S1. The error estimates were calculated directly from the fitting uncertainty, and are reported as the minimal and maximal intrinsic mean fluorescence lifetime values calculated from all lifetime component values and their amplitudes, using Supplementary Eq. 2, which have a reduced χ²-values within 95% confidence relative to the best-fit minimal reduced χ² value. Red line that is set at 1.44 ns highlights the limit below which fluorescence lifetimes drop in the presence of elevated concentrations of PEG or trehalose. Titration of NaCl or glycerol, leads to non-monotonic changes in τ which fluctuate around the typical fluorescence lifetime of mCherry.

groups inside mCitrine and eGFP? For PEG 6,000 and similar PEGs, the bulk refractive index reaches an asymptotically constant value of ~1.46 at an FVO as low as ~5%[49]. Therefore, it is clear that the fluorescence lifetime reductions in eGFP and mCitrine due to crowding occur on top of reductions potentially due to changes to the local refractive index. Indeed, it has been reported that increase of the bulk refractive index leads to reductions in the fluorescence lifetimes of eGFP[50] and mCitrine[51]. All of the physicochemical conditions that were tested induce significant enhancements of the bulk refractive index, including molar concentrations of NaCl. However, such high NaCl concentrations did not induce reductions in monomeric FPs fluorescence lifetimes (Fig.1, Supplementary Figs. 4–6). Nevertheless, even the high NaCl concentrations used here do not introduce increases in bulk refractive index[52] as high as those introduced by PEG[49]. In addition, the lifetime trends observed as a function of PEG are occurring while the bulk refractive index is expected to stay almost unchanged (i.e., PEG above 5% FVO)[49]. We can therefore conclude that the major reductions in fluorescence lifetimes above a given concentration or FVO threshold can be attributed to crowding. Importantly, we can conclude that in mCherry and mRFP, fluorescence lifetime reductions cannot be attributed to changes in viscosity or local refractive index, but rather to crowding alone. Similar to mCherry, all other FPs that were tested did not exhibit non-negligible spectral chromatic changes as a function of the different conditions (Supplementary Figs. 7–9).

**Mechanism of fluorescence lifetime reduction in FPs**
We move to examine the possibility that crowding of the FP from outside, above a given FVO value, could introduce degrees of freedom in the vicinity of the fluorophore group inside the FP, which could enhance the non-radiative de-excitation pathway. If internal degrees of freedom are introduced around the fluorophore, we expect it to exhibit more rotational degrees of freedom, relative to the β-barrel fold of the FP, as compared to the minimal degrees of freedom the fluorophore has relative to the β-barrel fold in buffer. Since fluorescence anisotropy decays track different types of rotational modes of a given fluorescent system through different timescales of different modes of depolarization, it can capture such effects[53]. This is mainly because the rotational correlation times of the fluorophore group relative to the β-barrel fold to which it is linked should be faster than the tumbling rotational correlation time of the whole FP.

We performed fluorescence anisotropy decay measurements of mCherry (Fig. 2a, b) in (1) 0% FVO PEG 6,000 (buffer only), (2) in 25% FVO PEG 6,000, just slightly below the 30% FVO limit, where only the increased bulk viscosity is expected to slow down motions and fluorescence lifetimes do not yet change (Figs. 1e), and (3) in 50% FVO PEG 6,000, where bulk viscosity is further increased, and also fluorescence lifetime decreases (Fig. 1e), yet potentially also the internal degrees of freedom of the fluorophore relative to the β-barrel fold of the FP (Fig. 2a, red, Fig. 1b). The fluorescence anisotropy decay was then calculated from the fluorescence decays polarized parallel and perpendicular to the excitation light plane of polarization (Supplementary Eq. (3)). The 10 ns window shown for the fluorescence anisotropy decay is the maximal time window that could still show the decay trend with bearable noise (Fig. 2b). Since the fluorophore system is constricted inside the FP, it is expected that mCherry in buffer will exhibit a fluorescence anisotropy decay with mostly a slow decaying component representing the slow tumbling of the whole FP, which is further slowed by the increase in viscosity due to the increase in PEG

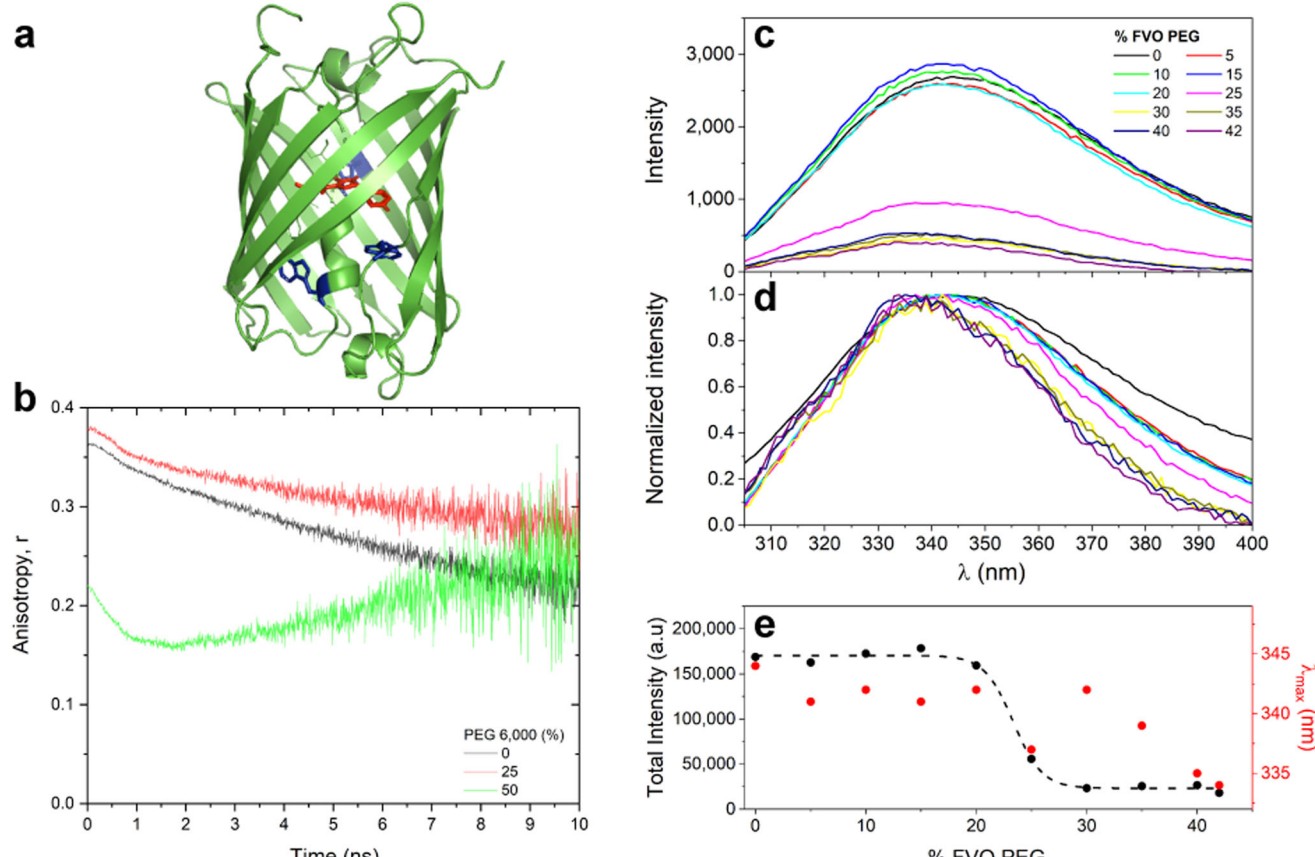

**Fig. 2 | Spectroscopic characterization of increase in degrees of freedom in the interior of mCherry. a** The fluorophore group of mCherry (red) and the three Trp residues of mCherry (blue) pointing towards the interior of mCherry (green). **b** Time-resolved fluorescence anisotropy of 100 nM mCherry show inner degrees of freedom of fluorophore increase. Shown are the fluorescence anisotropy decays of mCherry in the absence of PEG 6,000 (black) and in the presence of 25% (red) and 50% (green) FVO of PEG 6,000, where only above 30% FVO a rapidly decaying component shows up, on top of the overall slowdown due to increase in viscosity

and decrease in fluorescence lifetime. **c–e** Trp fluorescence spectra report on increase in FP interior degrees of freedom. The fluorescence spectra of the three Trp residues of 10 μM mCherry were recorded as a function of increasing FVO of PEG and are reported without normalization (**c**), and after normalization (**d**), and the area below the fluorescence spectra (**e**, black dots) and the wavelength at which fluorescence intensity is maximal (**e**, red dots) were reported. Note that the trend in the area below spectra was fit to a sigmoidal curve (**e**, dashed black line), with a best-fit midpoint at 23.3 ± 0.3% FVO of PEG.

concentration (Fig. 2b, black). On the other hand, the enhancement of internal degrees of freedom in the vicinity of the fluorophore due to crowding on the exterior of the FP introduces, aside from this slow protein tumbling process, another faster-decaying term (Fig. 2b, green). Note that this fast-decaying component appears at PEG > 30% FVO, above which the lifetime reduction starts to occur in mCherry (Fig. 1d–f). However, at 25% FVO, <30% cutoff, it seems that the effect of viscosity slowdown of protein tumbling is the only effect that occurs (Fig. 2b, red). We conclude that above the 30% FVO of PEG, fluorescence lifetime reduction is correlated with an enhancement of the rotational degrees of freedom of the fluorophore group relative to the FP β-barrel. Importantly, this change is most probably not due to Förster resonance energy transfer (FRET) between proximal (<10 nm) same FPs, better known as homoFRET, since the measurements were performed in bulk solution. Therefore, the observed appearance of a fast-decaying component in the fluorescence anisotropy decays could be due to the increase in the rotational degrees of freedom of the fluorophore group relative to the orientation of the β-barrel fold of mCherry. Moreover, according to the Perrin equation (Supplementary Eq. (4)), if there is a single mode of rotation, a decrease in fluorescence lifetimes should introduce an increase in fluorescence anisotropy, which is different from the observation. In fact, following the Perrin equation, the observed faster decay of anisotropy can be explained by a fast-decaying rotational correlation component.

Furthermore, to provide an atomistic explanation for fluorescence lifetime reduction caused due to macromolecular crowding by elevated FVO of PEG, we performed extensive MD simulations of mCherry in solution both in the absence of PEG and in the presence high %FVO of PEG, calculated to be 42% (e.g., >30% FVO PEG; Fig. 3a, b), for a combined total of 20 μs. The fluorophore participates in a network of fluctuating contacts with many residues in the protein (Supplementary Table 2). We refer to this network of residues as the fluorophore pocket. Additionally, we observe that water molecules are found within the pocket (Fig. 3c). The volume of the pocket fluctuates over the course of the simulation, and since water molecules are easier to displace than protein residues, the volume of the pocket can be used as a proxy for the flexibility of the fluorophore's environment. Therefore, to quantify the rigidity of the fluorophore's environment, we computed the volume of the fluorophore pocket, and show the distribution of pocket volumes for each system (Fig. 3d). The distribution of fluorophore pocket volumes in the presence of 42% FVO PEG is shifted significantly to larger volumes relative to the distribution in the absence of PEG, indicating that the fluorophore pocket has a larger volume on average under high levels of crowding. This shift in pocket volume suggests that interactions with PEG lead to a protein conformation in which there is more space in the vicinity of the fluorophore, and therefore room for more flexibility in the fluorophore's environment.

To further investigate the potential conformational basis of the pocket volume increase, we computed the average pairwise distances of the pocket residues (Fig. 3f). We observe that in the presence of 42% FVO of PEG, most pocket residues inside the β-barrel of mCherry have increased distance from each other, indicating a global expansion of pocket residues inside the β-barrel. Despite this difference in fluorophore pocket size, we observe no significant difference in hydrogen bonds between the fluorophore and the rest of the protein (Supplementary Fig. 10), consistent with the observation that absorption and emission spectra are unaffected by PEG[54]. Finally, the position of the central helix in mCherry is affected by the presence of PEG. Conformations in which the helix is farther from the β1-β5 and closer to the β7-β10 gap region are preferred (Fig. 3e, f). The β7-β10 gap is known to be a flexible part of mCherry[55]. Taken together, these results are consistent with the hypothesis that crowding mCherry from outside the β-barrel reduces the fluorescence lifetime of mCherry by inducing a conformational change that reduces the protein constraints on the fluorophore, increasing the rate of non-radiative de-excitation.

Following these results, we proposed to probe whether other residues that point towards the interior of the FP β-barrel, such as its three Trp residues (i.e., residues 58, 93, and 143), also experience the change in the interior of the FP due to crowding from its exterior. Note that while Trp93 and Trp143 are in the vicinity of the fluorophore (i.e., in direct contact 100% and 30% of the simulation snapshots, respectively), Trp 58 is not, yet is still part of the interior environment. We chose Trp since it can be selectively excited, and its fluorescence spectra can report on changes in their vicinities. The results of Trp fluorescence spectra measurements as a function of different FVO values of PEG report on an overall quenching transition occurring at 20-30% FVO PEG (Fig. 2c, Fig. 2e, black), with a gradual moderate blue shift of the wavelength of maximal fluorescence, $\lambda_{max}$, from 25% FVO and above (Fig. 2d, Fig. 2e, red). These results are correlated with the increase in the degrees of freedom in the FP interior. Indeed, the MD simulations report on a slight increase in solvent accessibility for Trp58 in the presence of high levels of PEG relative to when it is solely in water (Supplementary Fig. 11a–c). Additionally, the MD simulations reports on subtle $C_\alpha$-$C_\alpha$ distance changes between the Trp residues and their neighboring residues (Supplementary Fig. 11d–f), which could also explain the Trp fluorescence changes in terms of changing Trp immediate chemical environment.

**Utilizing FLIM to measure fluorescence lifetime reduction of FP precipitates in solution**
Reduction in the fluorescence lifetime of mCherry relative to its standard value can be used in FLIM to report on regions in a fluorescently imaged object with high local densities caused by crowding of at least 30% FVO (Fig. 1). However, we were not able to continue the assessment of fluorescence lifetimes in bulk solution >50% FVO of PEG. If so, how can we probe the continuation of the fluorescence lifetime reduction trend at FVO values of PEG > 50%? One way would be to measure fluorescence lifetimes in mCherry that precipitated, in solutions with PEG at given bulk concentrations. Increased levels of PEG and protein mixtures form precipitates, where the concentrations of the constituents are larger than in bulk solution[56–58]. We can therefore probe fluorescence lifetimes of FPs in these precipitates, and expect further reduction of fluorescence lifetimes in them relative to the fluorescence lifetimes in bulk solution. Therefore, we performed a controlled test on PEG-induced precipitates in solution, which were formed after mixing 100 nM mCherry with increasing PEG 6,000 FVO values (Fig. 4a, b). Then, we performed FLIM on the precipitates that accumulated on the surface of the coverslip (Fig. 4, c, d, e; Supplementary Movies 1, 2). In the absence of PEG and in the presence of PEG at 10% FVO, there was no sign of fluorescent precipitates accumulating at the surface of the coverslip. Fluorescent precipitates appeared at the

surface of the coverslip in the presence of PEG at 25% FVO, and at 50% FVO multiple μm-size precipitates were observed immediately upon preparation (Fig. 4a, b). The fluorescence images of precipitates exhibit: (1) irregular shapes (Figs. 4c), (2) different precipitates with different fluorescence lifetimes (Figs. 4c–e), (3) fluorescence lifetimes within precipitates are rather uniform per image frame (Figs. 4c), and (4) the fluorescence lifetimes within precipitates change between frames (Fig. 4e; Supplementary Movies 1, 2), and hence the precipitates exhibit dynamic changes in their densities. Importantly, the fluorescence lifetime values of mCherry in precipitates at bulk FVO > 25% were -1.10-1.25 ns, values that in bulk solution represent PEG at 40% FVO or higher (Fig. 1). Alternatively, at 25% FVO of PEG, in bulk solution fluorescence lifetime reduction does not occur (Fig. 1). However, because the concentrations in precipitates are higher than in bulk solution, we observed reduction in fluorescence lifetime even when we introduce 25% FVO into the bulk.

Taken together, when performing FLIM of monomeric FPs, such as mCherry, which has a typical fluorescence lifetime value of -1.44–1.50 ns, a decrease in the fluorescence lifetime can report on local densities or crowdedness per pixel of an image, under the assumption that crowdedness is at least above a given threshold. With this approach, we next set to characterize HP1α heterochromatin-associated condensates in differentiating pluripotent mouse embryonic stem cells (ESC) using FLIM.

**The phase of HP1α condensates in undifferentiated and early differentiated ESCs**
Significant changes in chromatin compaction and nucleosome plasticity occur during ESC differentiation[59], and nucleosome plasticity was shown to drive chromatin PS properties[60]. HP1 binding to heterochromatin and forming domains or condensates around it renders HP1 a crucial factor in heterochromatin structure. Despite several studies reporting different physical mechanisms underlying HP1α condensates[13,14,17,18,20,21,23–25], the basis of their formation in undifferentiated ESCs and during early differentiation is still obscured and has just started to be explored[28]. We, therefore, set out to study HP1α condensates before and after early ESC differentiation.

We performed FLIM of mCherry-fused HP1α, taking advantage of our previously-generated endogenously-tagged mCherry-HP1α fusion ESC clones[61,62], and report on the mCherry fluorescence lifetimes within these condensates. Despite the high mobility of HP1α throughout the nucleus[5,8], it forms stable condensed heterochromatin structures, which are actively renewed, as has been previously shown[18,19]. By using in-cell fluorescence correlation spectroscopy (FCS) measurements inside HP1α foci (Supplementary Fig. 12; Eqs. (5), (6))[63] and following photobleaching rate analyses (Supplementary Fig. 13; Eq. (7)), we confirm that HP1α retains its mobility inside the condensates, and therefore exists in a phase that allows translational mobility and is actively renewed. Next, we compared images of fluorescence intensity and FLIM of endogenously-tagged mCherry-HP1α ESC clones that grew either as undifferentiated ESCs or following 2 days of retinoic acid (RA)-induced differentiation, referred to here as early differentiated ESCs. While in the fluorescence images the intensities within condensates are rather homogeneous (Fig. 5a, d, g, j), the shapes of these condensates in undifferentiated ESCs tend to deviate from the circular and symmetrical shapes expected of LLPS condensates with an interior single liquid phase (Fig. 5a, d). However, after RA-induced early differentiation, the shapes of the observed HP1α foci are smaller and more circular (Fig. 5g, j). Additionally, while in undifferentiated ESCs there are only a few large condensates in each nucleus, after early differentiation cells display many smaller-sized condensates per nucleus, as previously reported[5,64].

The interior of a LLPS-induced condensate composed of an interior single liquid phase is expected to exhibit uniform density. Therefore, if mCherry-HP1α condensates formed through classic

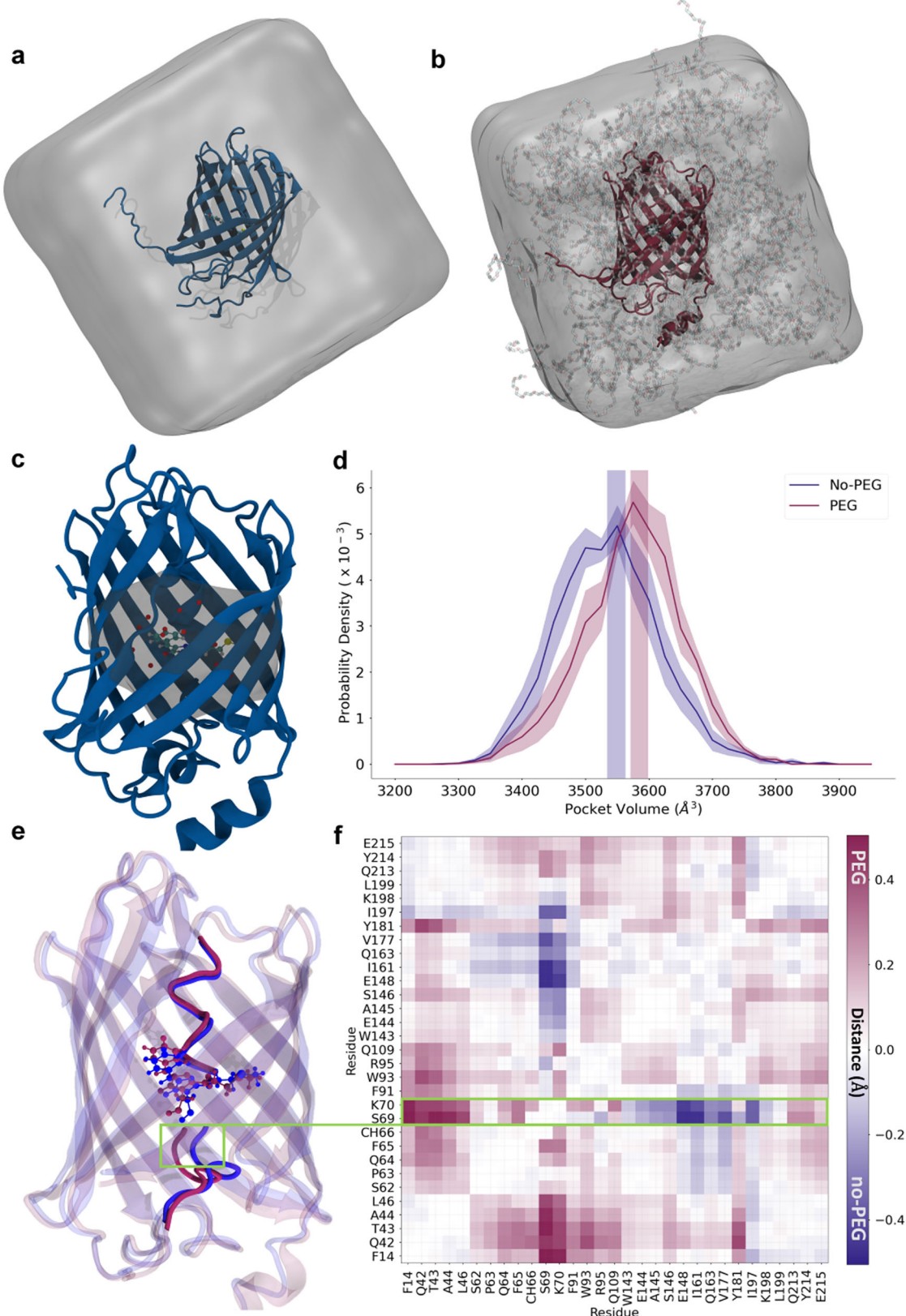

liquid-liquid phase separation induced by crowding of >30% FVO, we expect to observe a homogeneous (i.e., single narrow population) distribution of fluorescence lifetimes within all pixels of the condensate. On the contrary, our results show a varying range of fluorescence lifetimes in the interiors of HP1α condensates in undifferentiated ESCs (Fig. 5b, c, e, f). These condensates exhibited

sub-domains with lifetime values < 1.44 ns, below which >30% FVO PEG has shown to initiate the lifetime reduction trend (Fig. 1). Additionally, as can be judged by the FLIM images and the distribution of pixel-wise fluorescence lifetimes, mCherry-HP1α condensates in undifferentiated ESCs exhibit a heterogeneity of fluorescence lifetime values, wider than in early-differentiated ESCs (Fig. 5h, i, k, l). Therefore, FLIM of

**Fig. 3 | Conformational change of the fluorophore pocket in the presence of PEG. a, b** Simulated systems of mCherry in water (**a**) and mCherry in water and 42% FVO PEG (**b**). **c** mCherry with the fluorophore pocket (formed by the 30 residues shown in Table S2) shaded in. The fluorophore is shown in a ball and stick representation and water molecules are shown as red spheres. The structure is taken from simulations. **d** The distribution of the pocket volume for both the PEG and no-PEG systems. The PEG system is shown in plum and no-PEG is shown in navy blue.

Shaded regions indicate the standard error of the mean, treating each trajectory as an independent observation. **e** Representative 3D structures from the simulations showing helix conformations for the PEG (plum) and no-PEG (navy) systems. Residues S69 and K70 are highlighted in a green box. **f** The difference map of average fluorophore-pocket $C_\alpha$-$C_\alpha$ distances. Distances that are larger in the PEG system are positive and shown in plum, and distances that are larger in the no-PEG system are negative and shown in navy blue.

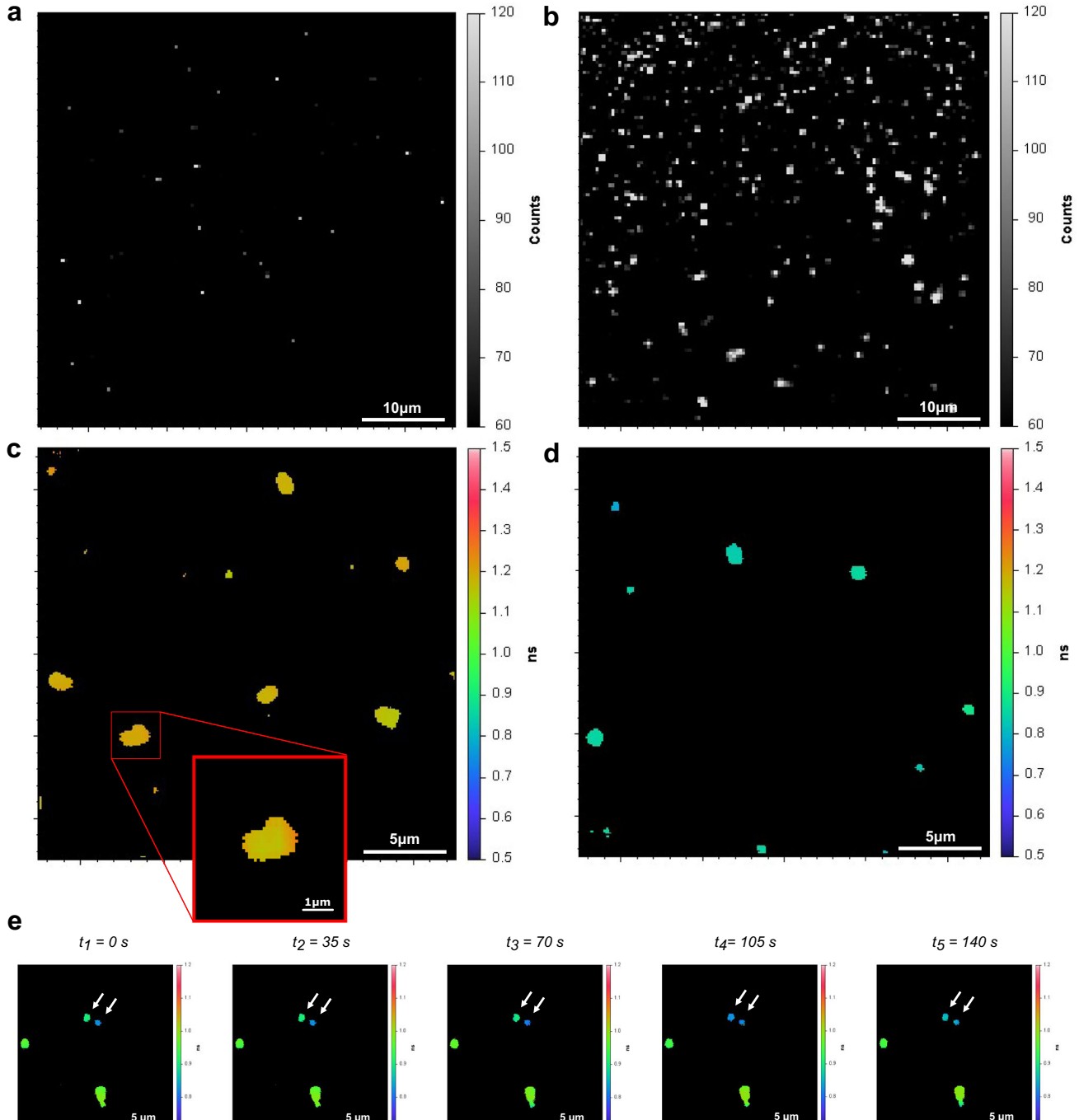

**Fig. 4 | Fluorescence intensity and lifetimes of PEG-induced in vitro mCherry precipitates in solution. a–d** Fluorescence lifetime (**a, c**) and intensity (**b, d**) images of mCherry PEG 6000-induced in vitro precipitates formed after adding mCherry at a total concentration of 100 nM in a solution containing 25% FVO (**a, b**) and 50% FVO (**c, d**) PEG 6000. **e** A few frames from a time series of fluorescence lifetime images of mCherry precipitates formed in a solution containing 50% PEG 6,000, with white arrows pointing on precipitates with visible changes in fluorescence lifetime over the course of 140 s, but with a homogeneity of lifetime values in each time point for each precipitate. Fluorescence lifetime colorbar scale from 0.5 and 1.5 ns.

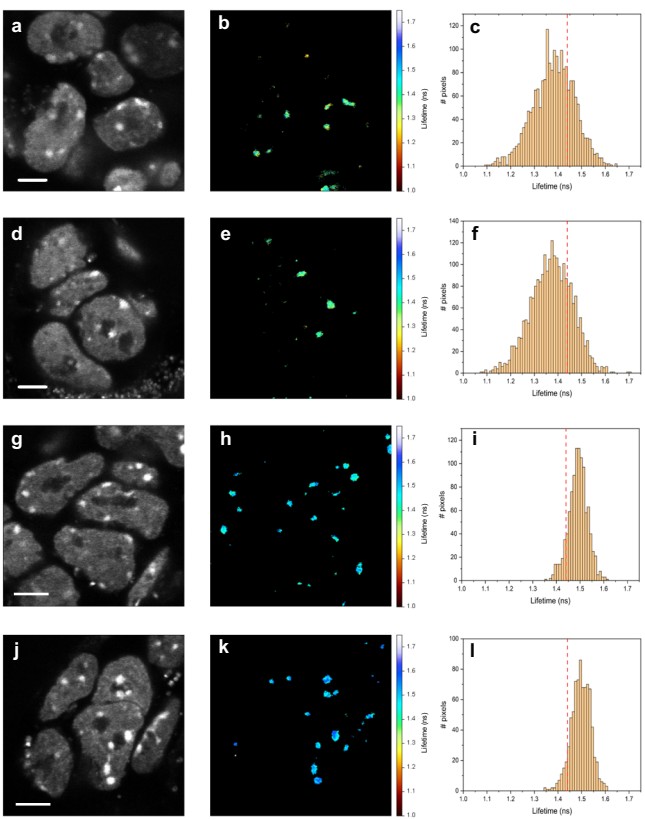

**Fig. 5 | fluorescence lifetimes of mCherry within mCherry-tagged histone protein 1α (HP1α) condensates are lower and more heterogeneous in undifferentiated than in early-differentiated mouse embryonic stem cells (ESCs). a–f** Undifferentiated ESCs. Two examples of undifferentiated ESCs with mCherry-HP1α condensates, fluorescence intensity images (**a, d**), fluorescence lifetime images (**b, e**) and the histogram of pixel fluorescence lifetimes (**c, f**). **g–l** Early-differentiated ESCs. Two examples of early-differentiated ESCs with mCherry-HP1α condensates, fluorescence intensity images (**g, j**), fluorescence lifetime images (**h, k**) and the histogram of pixel fluorescence lifetimes (**i, l**). A vertical red line shows the 1.44 ns lower boundary of mCherry fluorescence lifetime below which the crowdedness level is above 30% FVO (Fig. 1). Scale bar 2 μm. Fluorescence lifetime colorbar scale from 1.00 and 1.75 ns.

mCherry-HP1α shows that in undifferentiated ESCs it cannot be described by the classical description of LLPS condensates with an interior having a single liquid phase. Importantly, these condensates cannot be represented by a solid phase either, since they exhibited mCherry-HP1α diffusivities within them, which should be abolished in a solid phase (Supplementary Fig. 12). These results are in line with HP1α condensates in mouse ESCs that (i) adopt a mobile yet not a liquid single phase (e.g., gel-like phase, multi-phase LLPS), (ii) contain multiple sub-domains, each with its liquid phase and density, or (iii) cannot at all be described as a PS condensate.

Having established that undifferentiated ESCs display heterogeneous fluorescence lifetimes (Fig. 5b, c, e, f), as ESCs undergo RA-induced early differentiation, an increasing number of condensates shift towards fluorescence lifetimes longer than the 1.44 ns boundary, and with less heterogeneity of values across imaged condensates (Fig. 5h, i, k, l). These results indicate a transition upon ESC differentiation towards condensates that are less dense and with a more homogeneous density, hence a result that is more consistent with a LLPS-induced condensate with an interior single liquid phase upon differentiation. It is noteworthy that pixel-wise fluorescence lifetime values are at most weakly correlated with pixel-wise fluorescence intensity values (Supplementary Fig. 14).

One consequence of our results is that the densities in mCherry-HP1α condensates are larger in undifferentiated ESCs than they are after early differentiation. Importantly, since the elevated densities inside PS condensates are expected to be high enough to bring mCherry-HP1α constructs within 10 nm proximities, in which homo-FRET could occur, we also performed fluorescence anisotropy imaging. Our results show that in undifferentiated ESCs the fluorescence anisotropy values within mCherry-HP1α condensates are lower than in early-differentiated ESCs (Supplementary Fig. 15). In situations where no fluorescence lifetime changes are observed, these results would indicate higher degree of homoFRET in undifferentiated ESCs than in early-differentiated cells, serving as yet another indication of higher condensate densities in undifferentiated ESCs. Nevertheless, since we do observe fluorescence lifetime variations (Fig. 5), according to the Perrin equation (Supplementary Eq. (4)), homoFRET cannot be easily decoupled from fluorescence lifetime effects on the fluorescence anisotropy values. However, the magnitude of the difference in fluorescence anisotropy values between undifferentiated and early-differentiated ESCs is larger than expected from variability in fluorescence lifetime variations solely. Therefore, more homoFRET, hence denser condensates, occur in undifferentiated ESCs than after early differentiation. Interestingly, fluorescence anisotropy values in undifferentiated ESCs are more heterogeneous than in RA-induced early-differentiated ESCs, suggesting that the observed results are merely mirroring the fluorescence lifetime images shown above (Fig. 5).

While these results refer to the distribution of fluorescence lifetimes within HP1α condensates, examination of mean fluorescence lifetimes of overall HP1α condensates and their comparison between different condensates in multiple image acquisitions is necessary to explore the generality of the identified phase behavior and how it might distribute across multiple cells and image acquisitions. Therefore, we quantified the mean fluorescence lifetimes over all pixel values for each condensates over multiple condensates from multiple image acquisitions, and compared them between undifferentiated and early-differentiated ESCs (Fig. 6). The results report on the following trends: (1) in undifferentiated ESCs, the fluorescence lifetime values in HP1α condensates is distributed widely, a considerable fraction of which having values lower than the 1.44 ns boundary of the typical fluorescence lifetime value of mCherry (Fig. 6, dashed red line); (2) fluorescence lifetimes in HP1α condensates after early differentiation distribute in a narrow single population, with values predominantly above the 1.44 ns boundary of the typical fluorescence lifetime of mCherry. These results show that the mean densities of HP1α condensates indicate higher FVO than the threshold above which fluorescence lifetime reductions are observed (Fig. 1), but also point towards more heterogeneity in densities in undifferentiated ESCs than in early differentiated ones not only within but also between HP1α condensates.

In mouse ESCs, MSRs are highly-transcribed[65,66] and play a major role in defining heterochromatin structure and its regulation[28]. After transcription, MSRs remain in close proximity to chromocenters and have been shown to interact with heterochromatin-associated proteins, such as HP1α, and promote its phase separation[28]. Therefore, by transfecting undifferentiating ESCs with locked nucleic acid (LNA) oligonucleotide probes that deplete MSR transcripts with an efficiency of ~50% (gapmers)[28,67], thereby affecting their interactions with chromocenters, we expect mCherry-HP1α condensates to exhibit results closer to those observed after RA-induced early-differentiation, hence less heterogeneity in fluorescence lifetimes and higher fluorescence lifetime values. Reassuringly, undifferentiated ESCs treated with the MSR gapmers exhibited increased pixel-wise fluorescence lifetime values, similar to the typical mCherry lifetime values below 30% FVO (i.e., ~1.44–1.50 ns), and less heterogeneity when compared with non-targeting scrambled RNA control probes (Fig. 7a–l). HP1α condensates were less dense with lower density heterogeneity, very similar to the

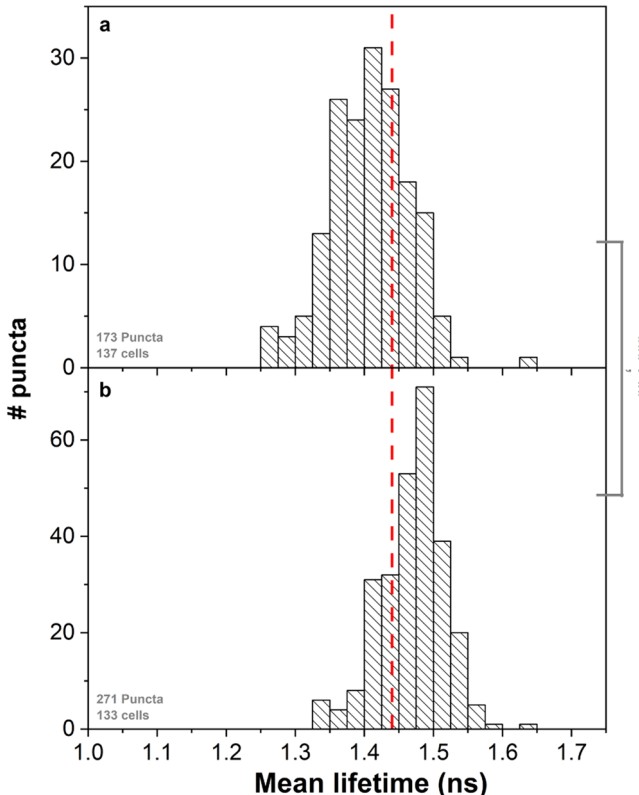

**Fig. 6 | Mean fluorescence lifetimes of mCherry-HP1α condensates from undifferentiated and early-differentiated ESCs.** Histograms show the distribution of mean fluorescence lifetimes of different mCherry-HP1α puncta in undifferentiated ESCs (**a**) and early-differentiated ESCs (**b**). Asterisks report on the *p*-value from two-sided *T*-test and from two-sided *F*-test, comparing means and variances of histograms. $p > 0.05$ n.s.; *$0.01 < p < 0.05$; **$0.001 < p < 0.01$; ***$p < 0.001$. A vertical red line shows the 1.44 ns lower boundary of mCherry fluorescence lifetime below which the crowdedness level is above 30% FVO (Fig. 1).

results in RA-induced early-differentiated ESCs (Fig. 5g–l). Performing the same transfection of the MSR-interacting LNA probes on RA-induced early-differentiated ESCs, revealed no significant effect on condensate formation and the low heterogeneity was maintained (Fig. 7m–r). We also examined the mean fluorescence lifetimes over all pixels of a condensate and compared these values between multiple HP1α condensates in multiple image acquisitions of undifferentiated ESCs and early-differentiating ESCs transfected with the MSR gapmers. ESCs treated with the gapmers showed increased mean fluorescence lifetime and reduced heterogeneity of lifetimes, similar to the results after RA induction. As a negative control, we performed similar analyses on ESCs transfected with the scrambled RNA probes and observed no change in the mean lifetimes or their heterogeneities between condensates (Supplementary Fig. 17), relative to the results in the absence of any transfection.

In summary, the results show that HP1α can accumulate in intranuclear condensates exhibiting diffusional mobility yet not necessarily single-liquid phase in undifferentiated ESCs, that gradually shift to a phase with features closer to a single-liquid phase upon induction of early differentiation conditions, and that reduced mCherry lifetimes can be used to estimate density properties of these bio-condensates to infer on their material properties.

## Discussion

In this work, we have shown that fluorescence lifetime reduction in FPs can serve as a reporter of increased local densities or crowding levels.

The results of the herein methodological developments warrant using FLIM to assess local densities of FP-fused proteins in cellular regions that experience high molecular densities, such as those found in small cellular compartments and in PS condensates.

There are many useful fluorescence-based approaches for studying the properties of PS condensates in the cell through measuring diffusivities, e.g., by in-cell FCS[68,69], or fluorescence recovery after photobleaching[25,70], through tracking fluorescence intensity images to capture condensate fusion and fission events[14] or by characterizing condensate circularity[14]. A key feature of LLPS condensates having an interior single liquid phase is the uniform density that is to be expected in the interior of a condensate, and hence heterogeneities in crowding are not expected. Therefore, a fluorescent reporter of local densities or crowdedness levels is necessary to characterize the actual physical phase of biomolecular condensates. Such fluorescent-based reporters of nanocompaction within bio-condensates were previously developed based on FRET between two different FPs (i.e. heteroFRET) fused through a crowding-sensitive protein[33,71–73]. However, FRET-based crowding probes require two spectrally-separated FPs, which reduce the spectral ranges available for imaging of other fluorescently-tagged targets in the cell. Additionally, since these FRET crowding sensors do not localize solely inside bio-condensates of choice, they might not necessarily provide enough contrast to distinguish between the interior and exterior of cellular condensates. It is noteworthy, that apart from fluorescence intensity scaling as a function of the number of FPs within an imaged pixel, it is also influenced by fluorescence quantum yields and lifetimes. Nevertheless, at least in our results, pixel-wise fluorescence intensity and lifetime values do not exhibit a dramatic correlation (Supplementary Fig. 14).

Another FRET-based approach to sense nanocompaction and high densities is the use of homoFRET, which relies on the transfer of excitation energy between two same fluorophores, owing to their proximity within 10 nm and the overlap- between their excitation and emission spectra. While in bio-condensates, it is plausible to expect FRET-relevant distances between nearby FP fluorophores, in homoFRET upon energy transfer, donor fluorescence decreases and in parallel, acceptor fluorescence increases. However, the same spectral region is covered for both donor and acceptor emission, which disables the use of homoFRET to be reported via changes in fluorescence intensities or lifetimes. Still, homoFRET can be reported via reduced fluorescence anisotropy values, since energy transfer does occur between same type fluorophores with different orientations in space, which further contributes to the depolarization of fluorescence, and hence to the reduced fluorescence anisotropy values. Yet, if fluorescence lifetimes of the fluorophores do not change, the sole reason for reductions in fluorescence anisotropy values would be the reduction in the rotational correlation time due to faster depolarization of fluorescence, as can also be understood from the Perrin equation (Supplementary Eq. (4)). Still, in our case, we observe also fluorescence lifetime variations, and therefore cannot report solely on changes of homoFRET in fluorescence anisotropy imaging. Interestingly, previous reports of sensing inner densities using fluorescence anisotropies have reported both changes in fluorescence anisotropies and lifetimes of FPs, and reported them to be caused by homoFRET, where the more homoFRET occurs, the more dense the FP-fused proteins are[32]. However, the fluorescence lifetime changes reported in that work could not be explained by a homoFRET mechanism and were not explained mechanistically. We suggest that this previous study of inner densities observed FLIM results similar to ours, although did not provide a mechanism to explain the FP lifetime variation dependence on high inner densities. Accordingly, we propose using FLIM to image proteins genetically fused to typical monomeric FPs that exhibit fluorescence lifetime reductions mostly in response to increased density and crowding above a given FVO threshold, to report on local densities inside PS condensates. Density heterogeneities within PS condensates

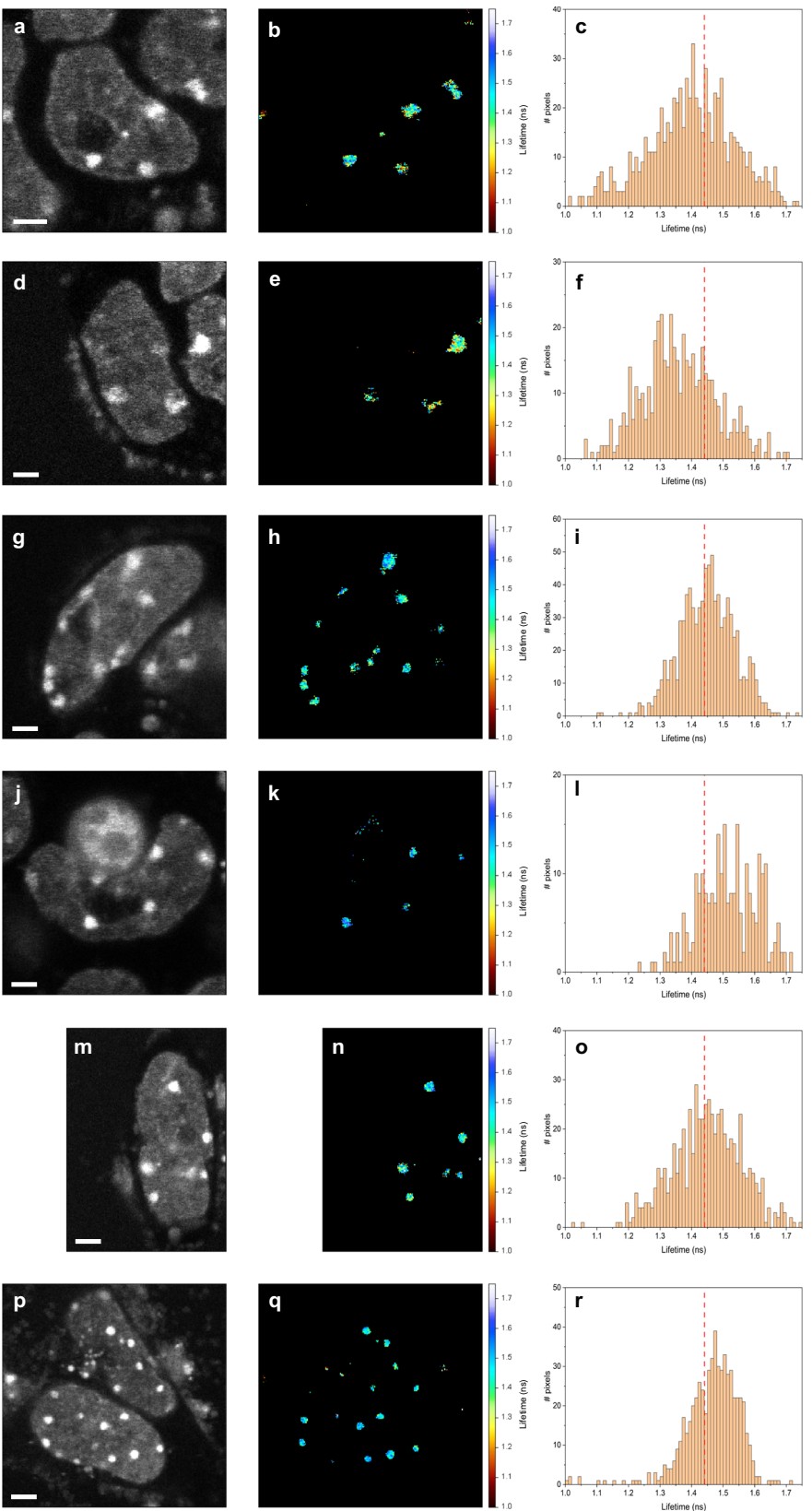

may imply a deviation from the single-liquid phase scenario, which together with findings from other methods could enable a more complete physicochemical characterization of the condensates under study. Importantly, since this approach is not specific to PS condensates, it may be used in other applications of cellular imaging with high levels of crowding.

Unlike mCherry and mRFP, our tests show that the fluorescence lifetimes of eGFP and mCitrine may be influenced not predominantly by crowding, but also by an increase in the local refractive index near the fluorophore inside the β-barrel fold. This effect is established in the Strickler-Berg relationship between the rate of radiative de-excitation of the fluorophore and the square of the refractive index in the vicinity

**Fig. 7 | Depletion of MSR transcripts in ESCs shifts HP1α condensates into a liquid-like form. a–f** Undifferentiated ESCs treated with an RNA scramble probe (negative control). Two examples of undifferentiated ESCs transfected with an RNA scramble probe, showing fluorescence intensity images (**a, d**), fluorescence lifetime images (**b, e**) and histograms of pixel-wise lifetime values (**c, f**). **g–l** Undifferentiated ESCs treated with LNA probes. Two examples of undifferentiated ESCs transfected with an LNA probe, showing fluorescence intensity images (**g, j**), fluorescence lifetime images (**h, k**) and histograms of pixel-wise lifetime values (**i, l**). **m–r** Early-differentiated ESCs treated with LNA probes. Two examples of early-differentiated ESCs transfected with LNA probe, showing fluorescence intensity images (**m, p**), fluorescence lifetime images (**n, q**) and histograms of pixel-wise lifetimes values (**o, r**). A vertical red line shows the 1.44 ns lower boundary of mCherry fluorescence lifetime below which the crowdedness level is above 30% FVO (Fig. 1). Scale bar 2 μm. Fluorescence lifetime colorbar scale from 1.00 and 1.75 ns.

of the fluorophore, hence depends inversely on the fluorescence lifetime. We have shown that this effect is observable in green-yellow FPs (e.g., eGFP, mCitrine; Supplementary Figs. 5, 6). We propose that this is mostly attributed to the previously reported lower structural stabilities of green-yellow FPs relative to red FPs (e.g., mRFP, mCherry), where this effect was negligible[74]. In short, we propose that the higher the structural stability of the β-barrel fold, the less the local refractive index in the microenvironment of the fluorophore will be influenced by changes in the bulk refractive index, which would then lower the impact of the Strickler-Berg effect on the radiative de-excitation pathway. Therefore, we propose to use fusions of mCherry, mRFP, or other monomeric FPs that exhibit fluorescence lifetime reductions mostly or solely due to crowding. Additional monomeric FPs designed to be highly stable have been reported in the literature[69] and are potential subjects for further investigations similar to those presented here.

We have demonstrated that crowding, acting from outside the FP, increases the available volume around the fluorophore, namely the pocket volume of the fluorophore inside the β-barrel fold. That, in turn, increases the degrees of freedom around the fluorophore, which lead to an enhancement of the non-radiative de-excitation pathway of the fluorophore, and hence to fluorescence lifetime reduction. These results could explain why >30% FVO of PEG the fluorophore exhibits rotational degrees of freedom relative to the β-barrel fold of the FP, which did not show up below that FVO value.

What could be the photophysical mechanism behind the increased rotational degrees of freedom around the fluorophore leading to the enhancement of its non-radiative de-excitation pathway? A closer look at the molecular structure of FP fluorophores could suggest that if they were not spatially restricted by the FP β-barrel fold, they would have undergone *cis-trans* photoisomerization, between isomers that emit photons upon de-excitation and isomerization intermediates, which do not emit photons upon de-excitation[75]. This way, since with no spatial constrictions photo-isomerization is very efficient[76–79], the overall fluorescence quantum yield would be low. Since the fluorophore is constricted within the β-barrel fold, it can be assumed that its photoisomerization is limited, and hence that its fluorescence quantum yield is optimized, if not maximized. Nevertheless, if the degrees of freedom around the fluorophore increase and allow some degree of fluorophore photoisomerization, this will decrease the fluorescence quantum yield, and hence also the fluorescence lifetime. Indeed, inhibition of *cis-trans* photoisomerization has been shown to occur in FP fluorophores when free in solution and out of the context of the β-barrel fold[75,80–82]. This characteristic of FPs is useful when mutationally designing FPs that undergo fluorescence blinking, by altering the spatial constriction of the fluorophore[82] for applications such as single-molecule localization microscopy[83,84].

Nevertheless, there could also be uses for FPs that exhibit high spatial restriction of their fluorophores to sense local environmental effects. Finally, while modulation of *cis-trans* photoisomerization is one way of facilitating fluorescence lifetime reductions upon enhancement of degrees of freedom in the interior of an FP, other photophysical mechanisms can induce fluorescence lifetime reduction upon such degrees of freedom increase. Additionally, other factors complicate the search for the underlying mechanism. For instance, FP fluorophores exhibit different ionic forms as a function of pH, which

could induce dramatic changes to their fluorescence characteristics[85]. Therefore, it is important to use FPs that have low pKa values, such as mCherry[86], and perhaps focus on imaging cellular compartments that do not exhibit large pH deviations from neutral values, such as in the nucleus. Importantly, while we got closer to a working mechanism of how mCherry might act as a fluorescence lifetime sensor of high density, the mechanism of how other monomeric FPs might act as density sensors will be the subject of a follow-up study, inspired by the results of this work.

To explore the utility of this method, we employed FLIM to assess the PS properties of HP1α in heterochromatin condensates in undifferentiated and early differentiating mouse ESCs. We used our endogenously-labeled mCherry-HP1α ESCs[61,62], to evaluate the fluorescence lifetimes in heterochromatin foci in both undifferentiated ESCs and in RA-treated ESCs for 2 days to capture the very early stages of differentiation. This non-invasive approach for monitoring protein condensation enabled us to identify heterochromatin protein reorganization processes inside of the foci in pluripotent and early differentiated cells. Previous work from us and others revealed increased chromatin protein dynamics in heterochromatin of pluripotent versus differentiated cells, where this dynamic association with heterochromatin is restricted[5–7,87–89], and several studies began to identify some of the underlying mechanisms, including specific factors[88], or chromatin-lamina interactions[6].

Here we tested whether the PS properties of heterochromatin may explain these previous observations. Intriguingly, we identified an early transition from a relatively heterogeneous distribution of fluorescence lifetimes in heterochromatin foci of undifferentiated ESCs to a more homogeneous state in the RA-induced early-differentiated ESCs, with an increasing number of foci entering longer and more homogeneous distribution of fluorescence lifetimes. The main difference we previously observed in HP1 dynamic association with chromatin between undifferentiated and early differentiated ESCs was the bleach depth[5], indicating that undifferentiated ESCs possess an additional sub-population of highly dynamic HP1 molecules, in addition to the one observed in early differentiated ESCs. This sub-population coincides with the increased density heterogeneity within the condensates we studied in undifferentiated ESCs, yet not in early differentiating cells. Our observations could be explained by a model of HP1 condensates with different liquid subdomains, each phase-separated from the other within the condensates themselves, and with its distinct dynamics. This would explain both the larger heterogeneity we observe within undifferentiated ESCs, and the transition from this heterogeneous multi-subdomain state into a more homogeneous liquid-like state in early differentiation. Indeed, several works have shown that HP1α condensates exhibit behavior that cannot be explained by a liquid phase, at least partially[14,18,90]. Recently, Novo et al. demonstrated that heterochromatin condensates, comprising MSR transcripts and HP1α, form in undifferentiated mouse ESCs[28]. Interestingly, these condensates exhibited features that can be partially explained as induced by LLPS towards an interior single liquid phase and others that cannot (e.g., irregular shapes, substantial immobile fraction), and hence coincide with the heterogeneities we report here. Interestingly, upon MSR transcript depletion, the heterochromatin condensates exhibited characteristics of homogeneous condensates. We have previously shown that ESC differentiation leads to the

reduction in such noncoding satellite repeat transcripts[66], and hence the Novo et al. findings[28] coincide also with our own regarding early differentiated mouse ESCs. To test this directly, we employed the same LNA gapmer probes used in the Novo et al. study to suppress MSR expression in ESCs. Since MSR transcripts promote HP1α phase separation[28], their suppression should eliminate at least one of the phase-separated HP1α populations in the heterogeneous undifferentiated state. Supporting this view, we observed a transition from a multi-phase state to a liquid-like state similar to the situation in early differentiating cells.

Our results on HP1α heterochromatin condensates in undifferentiated versus differentiated ESCs are in line with other recent reports of heterochromatin condensates[91,92]. Some of these works report on a liquid-like phase at the nanoscale with non-liquid behavior at the microscale[93], which could provide yet another explanation for our results regarding a behavior that is liquid-like from molecular diffusivities but not from the perspective of density heterogeneities at the nanoscale.

While we report a methodological concept, provide a working mechanism for it, and demonstrate its use, many questions are yet to be answered. On the FP side, how many other FPs can also exhibit the effects shown here by mCherry and mRFP? Could the reported crowding effects on internal fluorophore degrees of freedom be mutationally designed? On the application of FPs to study PS condensates, what further phase transitions could be studied with this approach combined with other experimental approaches? and most importantly, could the herein presented experimental approach assist in achieving a clear picture of PS condensates in cells? We envision these questions as well as others would be addressed in future studies building on this work.

## Methods

### Expression and purification of recombinant His6 tagged monomeric FPs

BL-21 competent cells were transformed using mCherry, eGFP, mRFP[94], and mCitrine plasmids acquired from addgene as a gift from Michael Davidson, Robert Campbell, Roger Tsien or Scott Gradia (addgene plasmid mCherry: #29722, eGFP: #54762, mRFP: #54667 and mCitrine: #29724). A single colony of transformed BL-21 cells was inoculated in 10 mL of LB media supplemented with 100 μg/mL Ampicillin or 50 μg/mL Kanamycin and incubated overnight at 37 °C and 250 rpm. The overnight culture was diluted (1:100, v:v) into a new autoclaved 2 L Erlenmeyer with 1 L LB media supplemented with 100 μg/mL of the appropriate antibiotic. Culture growth was monitored by periodically measuring $OD_{600}$. Once $OD_{600}$ reached 0.6 (~4 h after inoculation) culture was induced with 1 mL of 1 M IPTG (to final concentration of 1 mM) for mCherry and mCitrine or with 2% arabinose for eGFP and mRFP, and continued incubation for another 3 h. Cells were harvested by centrifugation at 6000 g and 4 °C for 10 min using 250 mL bottles in R12A3 rotor and Hitachi Koki himac CR22N high-speed refrigerated centrifuge. Pellets were stored at −80 °C. Pellets from 1 L cell media were resuspended in 50 mL of lysis buffer (100 mM sodium phosphate, 100 mM NaCl, pH 7.0), and lysed by ultra-sonication (10–12 cycles of 20 s pulses with 50 s intervals, at 60% amplitude). Cell debris was removed by centrifugation at 20,700 g, 4 °C for 30 min. HisTrap HP 1 mL was used to separate the his-tagged FPs from the supernatant using the AKTA system[95]. Fractions collected were run in SDS-PAGE to assess the purity of the protein. Pure fractions were dialyzed overnight in dialysis buffer (20 mM Na2HPO4, 50 mM NaCl, pH = 7.8) and stored in 50% glycerol at −20 °C.

### ESC culture and differentiation

ESCs were cultured on 0.2% gelatin-coated plates and grown on a feeder layer of mitomycin C-treated mouse primary embryonic fibroblasts (MEFs) and maintained in ESC media (DMEM, 10% FBS, 1000U of leukemia inhibitory factor [LIF], 2 mM L-glutamine, 1% nonessential amino acids, 50 units/ml penicillin, 50 μg/ml streptomycin and 0.1 mM β-mercaptoethanol). Cell cultures were maintained in a humidified atmosphere (5% CO2 at 37 °C). For RA-induced differentiation, ESCs were grown on gelatin-coated dishes without MEFs for up to 2 days in ESC medium without LIF and with 1 μM RA. For cell live imaging, cells were seed on an ibiTreat 8-well μ-Slides (#80826 ibidi, Munich, Germany) in appropriate cell culture medium and conditions.

### Depletion of major satellite repeat transcripts in mouse ESCs

LNA gapmers were provided by G. Almouzni[67]. Transfection was performed following the protocol presented by Novo et al.[28]. Briefly, cultured ESCs expressing mCherry-tagged HP1α were seeded on gelatin-coated ibiTreat 8-well μ-Slides in appropriate cell culture medium and conditions. Cells were transfected with LNA gapmer oligos (Exiqon) at a concentration of 100 nM together with an eGFP-expressing plasmid, using Mirus (TransIT-LT1) following the manufacturer's instructions. The LNA gapmer sequences are: LNA DNA gapmer control (gagaAAGTGTGACAagtg), LNA DNA gapmer Major Satellite 1 (acatCCACTTGACGActtg) and LNA DNA gapmer Major Satellite 2 (tattTCACGTCCTAAagtg). The LNA gapmer transfection was repeated after 48 h to ensure robustness.

### Experimental setup for FLIM, fluorescence lifetime, FCS and time-resolved fluorescence anisotropy measurements

We performed fluorescence lifetime imaging (FLIM) and in-cell fluorescence correlation spectroscopy (FCS) using a confocal-based microscopy setup (ISS™, Champaign, IL, USA) assembled on top of an Olympus IX73 inverted microscope stand (Olympus, Tokyo, Japan). We used 532 ± 1 nm pulsed picosecond fiber laser (FL-532-PICO, CNI, China; pulse width of 100 ps FWHM, operating at 20 MHz repetition rate and 100 μW, measured at the back aperture of the objective lens) for exciting mCherry-HP1α. The laser beam passes through a polarization-maintaining optical fiber (P1-405BPM-FC-Custom, with specifications similar to those of PM-S405-XP, Thorlabs, Newton, NJ, USA) and after passing through a collimating lens (AC080-016-A-ML, Thorlabs), the beam is further shaped by a linear polarizer (DPM-100-VIS, Meadowlark Optics, Frederick, CO, USA) and a halfwave plate (WPMP2-20(OD)-BB 550 nm, Karl Lambrecht Corp., Chicago, IL, USA). A major dichroic mirror with high reflectivity at 532 and 640 nm (ZT532/640rpc, Chroma, Bellows Falls, Vermont, USA), for mCherry, and at 488 and 640 nm (ZET488/640 m, Chroma, Bellows Falls, Vermont, USA), for GFP, reflect the light to the optical path through galvo-scanning mirrors (6215H XY, Novanta Corp., Boston, MA, USA) and scan lens (30 mm Dia. x 50 mm FL, VIS-NIR Coated, Achromatic Lens, Edmund Optics, Barrington, NJ, USA; both used to acquire the scanned image), and then into the side port of the microscope body through its tube lens, positioning it at the back aperture of a high numerical aperture (NA) super apochromatic objective (UPLSAPO100XO, 100X, NA = 1.4, oil immersion, Olympus), which focuses the light onto a small effective excitation volume, positioned within the sample chamber (μ-Slide 8 Well high Glass Bottom, Ibidi, Gräfelfing, GmbH). Scattered light returns in the excitation path, and a fraction of it is imaged on a CCD camera, used z-positioning, using Airy ring pattern visualization. Fluorescence from the sample is collected through the same objective, is transmitted through the major dichroics and is focused with an achromatic lens (25 mm Dia. x 100 mm FL, VIS-NIR Coated, Edmund Optics) onto a 100 μm diameter pinhole (variable pinhole, motorized, tunable from 20 μm to 1 mm, custom made by ISS™), and then re-collimated with another achromatic lens (AC254-060-A, Thorlabs). Fluorescence is then further cleaned using a 615/24 nm, for mCherry, or 510/20 nm, for eGFP, single band bandpass filter (FF01-615/24-25, or FF01-510/20-25, Semrock, Rochester, NY, USA). Fluorescence was collected using one hybrid PMT, routed to a TCSPC card (SPC 150 N,

Becker & Hickl, GmbH). Images were attained by using a laser scanning module (LSM), in which a 3-axis DAC module (custom made by ISS™) synchronized the data acquisition and control over the x & y galvo-scanning mirrors, which assisted in bringing the effective excitation volume to different positions to acquire pixel data per a given z layer. For in vitro FLIM measurements, lifetime images were attained by tail fitting the mCherry fluorescence decays of each acquire pixel, if it had at least 30 photons, where the scanning conditions are pixel dwell time −0.1 ms; number of pixels 256 × 256; field of view area −50 × 50 5 μm; number of acquired frames per image−5. Data acquisition is performed using the VistaVision software (version 4.2.095, 64-bit, ISSTM) in the time-tagged time-resolved (TTTR) file format.

In FLIM, lifetime images were attained by tail fitting the mCherry fluorescence decays of each acquire pixel, if it had at least 100 photons. Tail fitting was performed using a mono-exponential function, and the intrinsic mean lifetimes were reported. For retrieving mean fluorescence lifetime values of bulk fluorescence decays of recombinant mCherry, we performed a fitting procedure, with equation (1) and (2) (see in Supplementary), as explained in the text, using a Fortran-based software from Haas and colleagues[96,97]. Measurements were performed in PBS buffer (50 mM $NaH_2PO_4$, 100 mM NaCl, 30 mM KCl, pH = 7.0) at 100 nM protein concentration.

Time-resolved fluorescence anisotropy measurements (fluorescence anisotropy decays) were performed by adding a polarizing beam-splitter between two detectors, where one detector reports on fluorescence in a polarization parallel to that of the excitation polarization, and the other perpendicular to the excitation polarization. It is noteworthy that in the setup, the excitation is conditioned so that it reaches the sample with a vertical polarization. Then, fluorescence decays in the polarization parallel and perpendicular to the excitation polarization were recorded. The fluorescence anisotropy decay was calculated out of the fluorescence decays, after (1) correcting the perpendicularly polarized decay relative to the parallel polarized decay with a G factor, and after (2) one decay was shifted relative to the other by the time interval between them, which is due to the slightly different light path lengths for the two detectors. Overall, the anisotropy decay was calculated as in Supplementary Eq. (3). These measurements were performed on mCherry at 100 nM concentration in the PBS buffer mentioned above. Using free dye, we measured the expected G factor which was used in calculating the anisotropy. For further explanations, see SI.

In-cell FCS was performed by positioning the laser focus at a given point within a ROI and recording the fluorescence trajectory at this point for 2 min. The fluorescence trajectories typically exhibited a monotonic decay of fluorescence due to photobleaching. Next, the fluorescence autocorrelation function was calculated from the acquired fluorescence trajectories (Supplementary Eq. (4)), as previously shown[68,69]. The fluorescence autocorrelation functions were fitted to a model that includes two modes of translational 3D diffusion, with different fractions (Supplementary Eq. (5); see Kim, Heinze & Schwille[69]). This model was chosen after we found that it improved curve fitting relative to the fitting of a model with a single 3D diffusion species. Since one diffusion time component included values that do not make sense for measuring 3D diffusion in the cell (i.e., few ms or faster that equate to free organic dyes), we report the diffusion times of the slow component, which are also representative of the fluorescence autocorrelation functions (see Supplementary Fig. 12b).

To characterize the fluorescence photobleaching rates in the in-cell fluorescence trajectories acquired, we fitted them to a bi-exponential decay model (Supplementary Eq. (6)), as is performed in other similar studies[98,99] (Supplementary Fig. 13). Importantly, the timescale of fast and slow photobleaching are slower than the timescales reported in the FCS fluorescence autocorrelation curves. Therefore, it can be assumed that the FCS analyses were of diffusion-driven fluctuations that occur faster than photobleaching rates, and hence are not influenced by the photobleaching. Therefore, the FCS analyses results for the diffusion times are reliable. On the other hand, due to the ever-decreasing fluorescence, and hence and ever decreasing number of fluorescent molecules in the laser focus, the FCS analyses results for the mean number of molecules free parameter are unreliable. In addition, the results of the analyses of photobleaching rates carry additional information regarding diffusion: while slowly diffusing species stay for longer dwell times within the laser focus, and hence exhibit a higher probability to undergo photobleaching, rapidly diffusing species stay short dwell times within the laser focus, and hence exhibit lower probabilities to undergo photobleaching. In short, while the fast-decaying component of the photobleaching can report on slowly diffusing species, the slow decaying component can report on faster diffusing species. We therefore report the results of the photobleaching decay analyses (Supplementary Fig. 13). Analysis of the FCS data and result representation was performed using the OriginLab Origin software version 2022b.

## Analyses of mean fluorescence lifetime of HP1α condensates from pixels of FLIM data

Analysis was performed using ISS VistaVision version 4.2. Lifetime images were attained as described above. From the lifetime images, regions of interest (ROI) including most all pixels of HP1α condensates were manually selected. For each ROI, mean fluorescence lifetime over all of the condensate pixels, and the stranded deviation were calculated by averaging the fluorescence lifetime over all pixel-wise values, while making sure to ignore all 0 values (i.e., values outside the condensate for which fluorescence lifetime was not estimated due to lack of sufficient number of photons in these pixels). The mean fluorescence lifetimes of all condensates from all cells of a given condition were plotted in histograms showing distribution of condensate mean fluorescence lifetimes.

## Statistical comparisons between fluorescence lifetime distributions

In this work, histograms of mean fluorescence lifetimes of different condensates were compared on the basis of their mean and variance values using two sided T-test and two-sided F-test. The p-values of these comparisons are reported in the results using the following connotations: $p > 0.05$ n.s.; *$0.01 < p < 0.05$; **$0.001 < p < 0.01$; ***$p < 0.001$.

## Fluorescence anisotropy images

Fluorescence anisotropy images were acquired as described above (see *Time-resolved fluorescence anisotropy measurements*) in imaging mode, on both undifferentiating and early-differentiating ESCs, using a polarized beam splitter to report on fluorescence in two channels, one in a polarization parallel to that of the excitation polarization, and the other perpendicular to the excitation polarization. Analysis was performed using ISS VistaVision version 4.2. To report on fluorescence anisotropy of HP1α condensates only, we first set different intensity thresholds on both parallel and perpendicular polarized fluorescence channels, so that both thresholds would lead to showing the condensates with their edges in both channels almost matching. Next, we calculated the fluorescence anisotropy following the equation presented above (Supplementary Eq. 3), however for fluorescence intensities rather than for fluorescence decays. Due to higher intensity in the parallel channel relative to the perpendicular channel it is difficult to yield a high correlation in the position of foci, which leads to pixels on the outer edge of the foci to be visible in the parallel channel but not in the perpendicular. This, in turn, leads to a few pixels on the outer edge of the foci to appear as if they have anisotropy values higher than the maximum possible theoretical value of 0.4. Therefore, we filtered out any value that was >0.4 (Supplementary Fig. 15).

## Trp fluorescence spectra

Trp fluorescence spectra of mCherry in the absence and presence of different FVO levels of PEG were recorded using a spectrofluorometer (Jasco FP8200ST, Japan), scanning emission wavelengths in the range 305–400 nm and focusing on excitation at $\lambda = 295$ nm.

## Molecular dynamics simulations

**System setup and simulation protocol.** The crystal structure of mCherry was taken from the Protein Data Bank (PDB ID 2H5Q)[34]. Missing atoms were added with CHARMM-GUI PDB-reader tool[100]. Missing residues at the termini and a six residue histidine-tag at the C-terminus were added with CHIMERA version 1.15[101] and their structures were modeled with the MODELLER loop refinement tool[102]. Five structures were generated and the one with the lowest discrete optimized protein energy score[103] was used as the initial structure for MD simulations.

All simulations were performed in GROMACS version 2022.2[104,105]. The mCherry anionic fluorophore parameters[106] were converted to GROMACS format using the charmm2gmx python script with some modifications for Python3 compatibility. The CHARMM36m force field[107] was used with the CHARMM-modified TIP3P water model[108]. The N- and C-termini were protonated in the charged state and all titratable residues were simulated in their standard states for pH 7. For the simulations of the protein in water, the protein was positioned in a triclinic box with 10 Å distance to all box edges. Periodic boundary conditions were used. Water molecules were added to the simulation box and neutralizing $Na^+$ and $Cl^-$ ions were added at a concentration of 139.7 mM. Energy minimization was performed using the steepest descent method. The system was equilibrated for 10 ns in the NVT ensemble with 1,000 kJ/mol position restraints on all protein heavy atoms. The v-rescale thermostat[109] was used and velocities were generated from a Maxwell distribution. This was followed by a 10 ns constant pressure simulation using the Berendsen barostat[110] and the v-rescale thermostat[109], without changing the position restraints. The Parrinello-Rahman barostat[111] and the v-rescale thermostat[109] were used for all subsequent simulations. Next, the system was equilibrated for 20 ns in the NPT ensemble without changing the position restraints. The system was then equilibrated without any position restraints for 100 ns. Finally, five replicate production simulations were run for 2 μs in the NPT ensemble. A time step of 2 fs was used for all simulations. The bonds involving hydrogen atoms were constrained using the LINCS algorithm[112] and water molecules were constrained using the SETTLE algorithm[113]. Short-range electrostatic interactions and van der Waals interactions were computed with a cutoff of 12 Å. Long range electrostatics were computed using Particle Mesh Ewald summation[114] with 16 Å grid spacing and fourth order interpolation. The Verlet cutoff scheme was used for neighbor searching. The temperature was maintained at 298 °K, and the pressure was maintained at 1 bar.

A PEG-water system with 200 PEG 28-mer molecules and 6184 water molecules was set up using CHARMM-GUI[100] polymer-builder tool. The energy of the system was minimized using the steepest descent method. The initial equilibration of the system was carried out using the MD parameter file provided by CHARMM-GUI, which specified the Nose-Hoover thermostat[115]. This equilibration was run in the NVT ensemble for 250 ps, followed by a 1 ns equilibration simulation in the NPT ensemble using the Parrinello-Rahman barostat[111]. This was followed by a 100 ns equilibration simulation in the NPT ensemble using the v-rescale[109] thermostat and the Parrinello-Rahman barostat[111]. Six random conformations of PEG were selected from the final frame of this simulation and were used for setting up the protein-water-PEG system. The pre-equilibrated structure of mCherry from the simulations in water was used for initial setup of the PEG system. All water molecules and ions were removed. PEG molecules were inserted into the simulation box using the GROMACS tool gmx insert-molecules.

Water molecules were then added to the system along with neutralizing $Na^+$ and $Cl^-$ ions at a concentration of 139.7 mM, for a final PEG concentration of 42% FVO. The energy of the system was minimized using the steepest descent method. The simulations then followed the same protocol as in the system without PEG (see above).

**Analysis of simulation trajectories.** The first quarter of each simulation was treated as an equilibration period and omitted from all analyses on the basis of radius of gyration, number of hydrogen bonds, and root mean squared deviation. All coordinate-based trajectory analysis was performed using the MDAnalysis module version 2.2.0[116] for Python version 3.7. Residues were classified as fluorophore pocket residues if they formed contacts with the fluorophore in >0.33% of all simulation snapshots in each system. Residues were classified as Trp pocket residues if they formed contacts with each Trp in at least 0.33% of simulation snapshots. Two residues were considered to be in contact if any of their atoms were within a distance cutoff of 3.5 Å. Pocket volumes were defined as the convex hull volume of all $C_\alpha$ atoms in the pocket residues. The convex hull was determined with the Open3D module version 0.15.0[117] for Python version 3.7 after extracting the atomic coordinates using MDAnalysis[116]. Hydrogen bonds were computed with MDAnalysis version 2.2.0 using a distance cutoff of 3.5 Å and an angle cutoff of 150°. Uncertainty is reported as the standard error of the mean, with each independent trajectory being treated as an independent measurement.

## Statistics and reproducibility

All experiments performed, which include statistical analyses, were performed in biological repeats and results were validated. To study the fluorescence intensity and lifetimes of PEG-induced in vitro mCherry precipitates (Fig. 4), the experiment was repeated $n = 2$ per each of the indicated PEG concentrations and results were always similar. To study the fluorescence lifetimes of mCherry within mCherry-tagged HP1α condensates (Fig. 5), the experiments were performed in $n = 3$ biological repeats (different cell cultures), with multiple technical repeats (different frames of the same cell culture) in each biological repeat, and no significant deviation was observed in the results between both types of repeats. Studying the effect of depletion of MSR transcripts in ESCs (Fig. 7), we performed experiments with $n \geq 3$ biological repeats (different cell cultures), with multiple technical repeats in each biological repeat (different frames of the same cell culture), and no significant deviation was observed in the results between both types of repeats. Studying the diffusivities within HP1α condensates using in-cell FCS (Supplementary Fig. 12), performing $n \geq 2$ biological repeats (different cell cultures), with multiple technical repeats in each biological repeat (different positions within different frames of the same cell culture), and both biological repeats show similar results in all tested conditions. Performing fluorescence intensity and lifetime correlation, as well as fluorescence anisotropy tests (Supplementary Figs. 14 and 15) in $n = 2$ biological repeats (biologically independent samples) we observed similar results.

## Reporting summary

Further information on research design is available in the Nature Portfolio Reporting Summary linked to this article.

# Data availability

The raw fluorescence spectra, decays, and anisotropy decays, as well as the fluorescence decay fitting results generated in this study have been deposited in a publicly available Zenodo repository [https://doi.org/10.5281/zenodo.7964824]. This repository also includes the raw in-cell fluorescence intensity trajectories, their autocorrelation curves and their analyses, as well as the raw FLIM data and their analyses. Finally, this repository also includes the full MD simulation trajectory and its analyses results. A description of the data available in the mentioned

repository and generated in this study, is provided in a Source Data file. The crystal structure of mCherry serving as the bases for MD simulations was taken from the Protein Data Bank (PDB ID 2H5Q). Source data are provided with this paper.

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

## Acknowledgements

We thank Genevieve Almouzni for kindly providing the gapmers for MSRs. This research was supported by the Israel Science Foundation (grants 556/22 and 3565/20 to E.L.), the European Union's Horizon Europe Research and Innovation Programme under the EIC Pathfinder-Open grant agreement #101099654 (*RT-SuperES* to E.M.), the Natural Sciences and Engineering Research Council of Canada (NSERC) Discovery Grant (2018-06408 to S.R.), Calcul Québec (calculquebec.ca) for computational resources (to S.R.), and the Digital Research Alliance of Canada (alliance can .ca; to S.R.). E.M. is the Arthur Gutterman Family chair for Stem Cell Research.

## Author contributions
K.J. and S.D. produced and purified recombinant monomeric FPs, K.J. performed fluorescence spectra and lifetime characterizations and FLIM of monomeric FP precipitates for all recombinant monomeric FPs, as well as mCherry fluorescence anisotropy decays and Trp fluorescence spectra, both acquisition and analyses, L.H.-N. and S.R. designed, performed and analyzed all atom explicit solvent molecular dynamics simulations, J.O.V. performed ESC-related experiments, K.J., J.O.V., P.S.L.L., S.B. and P.D. performed confocal microscopy acquisitions of ESCs. K.J., S.B. and E.L. analyzed the fluorescence lifetimes of FLIM images and P.S.L.L. and E.L. analyzed the in-cell FCS curves and the accompanying photobleaching curves. E.M. and E.L. supervised the experiments. K.J., L.H.-N., S.R., E.M. and E.L. wrote the paper and all co-authors assisted in refining it.

## Competing interests
All authors declare no competing interests.
