## [Peer Review File · Nature Communications]

REVIEWER COMMENTS

Reviewer #1 (Remarks to the Author):

The authors present an excellently written and comprehensive study of the application of fluorescent proteins as sensors of phase transitions in the nucleus. They present fluorescence lifetime measurements as a valuable tool to study phase separations/transitions and changes in molecular crowding during chromatin activation in the differentiation process of ESC. In the first part, the authors perform all the necessary control studies to characterize and differentiate the effects of refractive index and molecular crowding on the fluorescence lifetime of fluorescent proteins. Interestingly, mCherry exhibits a different behavior than mCitrine and EGFP.

In the second part, experimental data and molecular modeling are used to convincingly demonstrate how a change in molecular density in the environment can contribute to a faster de-excitation of the excited fluorophore.

Eventually, in the third part, FLIM was employed to determine differences in heterochromatin foci in the nucleus of undifferentiated and retinal-stimulated ESC. The authors found differences in the fluorescent lifetime of mCherry-HP1 α distribution, with a less heterogeneous distribution of tau in differentiated cells. They suggest that HP1 is found in foci that are characterized by different subdomains within the foci and distinct molecular dynamics.

The study is well designed and conclusive.

The demonstration and professional introduction of the FLIM technique allows its application beyond the specific study and is of general interest since it advertises FLIM as a method for imaging and identifying phase transitions in living cells.

We suggest publication after a minor revision.

Minor:

Please comment on the time course of the anisotropy decrease at high PEG (Fig. 2b). Do you expect a further increase in anisotropy after 20 μ s? This would correlate with your proposed greater freedom.

Fig. 4: Please clarify if the precipitate environment is still wet or if it has dried out.

Page 9, last paragraph before new subtitle (fluorescence lifetimes lower than their standard values can report on local densities or crowdedness per pixel of an image), please include measured tau values in the text showing the difference described.

Mandatory: the authors mention a study (Novo et al.) showing that heterochromatin condensates have the characteristics of homogeneous condensates after depletion of MSR transcripts. Determining the tau value of the probe under these conditions would be a valuable contribution and would support your interpretation of the data.

Reviewer #2 (Remarks to the Author):

- What are the noteworthy results?
- Will the work be of significance to the field and related fields? How does it compare to the established literature? If the work is not original, please provide relevant references.
- Does the work support the conclusions and claims, or is additional evidence needed?
- Are there any flaws in the data analysis, interpretation and conclusions? Do these prohibit publication or require revision?
- Is the methodology sound? Does the work meet the expected standards in your field?
- Is there enough detail provided in the methods for the work to be reproduced?

The authors use fluorophore lifetime as an assay to address changes in density within biological systems. The authors begin by showing in vitro that the fluorescence lifetime of mCherry and other fluorophores are decreased by molecules that increase molecular crowding. They attribute this to an increase in the degrees of freedom of the fluorophore within the fluorescent protein based on anisotropy measurements and molecular simulations. They then use this to show changes in the densities of HP1alpha condensates in undifferentiated and early differentiated ESCs.

Main comments:

Overall, the use of FLIM to study heterogeneities within condensates is exciting. FLIM is a sensitive tool for monitoring regions within live cells and so it would indeed be nice to use it to report on heterogeneities in density with condensates, especially since one could combine this with other imaging modalities e.g. to show that RNA within heterochromatin condensates drive these inhomogeneities. The biological conclusion is also exciting in principle with the idea that HP1 condensates go from heterogeneous condensates to a homogeneous state as ESCs start to differentiate. A recent study they reference highlighted changes in RNA within ESC HP1alpha condensates that could be driving these differences.

Sadly, enthusiasm for the paper was somewhat reduced by some of the issues described below:

1. I found the intro confusing as it started with the basics of a fluorophore before we realise the authors are looking for a way to use fluorescent lifetimes as a density readout. It may be better to describe the alternate approaches for studying density, then discuss how FLIM is being used to address this and then discuss fluorophores and their lifetime, introducing papers using this approach and explaining how this paper contributes to the field.

2. The authors use molecular simulations to show that PEG is altering protein conformation (Fig 3) and that this is consistent with anisotropy changes (Fig 2) and lifetime (Fig 1). These experiments are very clear and well done. I also like the discussion making clear that some of the other fluorophores in Supplementary are affected by glycerol too so there may be other mechanisms at play for these fluorophores. The idea of using fluorophore lifetime or anisotropy to study the density of fluorophore-tagged proteins is not novel per se (PMID: 26133241; PMID: 35298090; PMID: 24703307). There have even been attempts to improve mCherry quantum yield through mutations that affect lifetime (PMID: 35709514). Having said that, previous studies have focussed on how FRET or homo-FRET affects lifetime/anisotropy and taken advantage of these changes to readout local density i.e. homo-FRET reduces fluorophore lifetime and affects anisotropy measurements. PMID: 26133241, for example, uses changes in the lifetime of mCherry, one of the proteins studied here, but attributes the change in lifetime to homo-FRET. It seems that the contribution of this paper is to show that PEG and crowding can itself affect FP conformation and therefore fluorophore anisotropy. It may therefore be important for the authors to explain how their results relate to previously proposed homoFRET based changes. Does PEG affect fluorophore intensity/quantum yield or alter FRET efficiency?

3. Next, they use Fig1-3 results as a justification for FLIM in Fig 4-6. However, it is not clear if the same mechanism of crowding is affecting the lifetime measurements inside living cells. Possible experiments (not all are required) that could make this more convincing include correlations between HP1 condensate intensity and lifetime, live-cell measurements of anisotropy or orthogonal evidence that HP1 is more dense in ES cells (does the intensity within HP1 foci relate to the lower lifetime measurements?).

4. Finally, the authors conclude from Figure 5-6 that HP1 condensates are heterogeneous in ESCs prior to differentiation. Although exciting if true, the authors show data from condensates within 5-15 cells. It is unlikely that this accounts for cell cycle which is known to affect condensate formation so at least 20 cells would be needed to be sure of this result. More cells may allow multi-Gaussian fitting and BIC analysis to demonstrate that there is more than one peak to the undifferentiated inside foci. It should be noted in the discussion that cell cycle will change during differentiation which could also affect the results.

5. Finally, there is a feeling that there are two papers in one here and the authors should make an attempt to better connect the two halves. The first half delves into the mechanism of lifetime changes when mCherry density is increased and then the second half is about using FLIM to show

heterogeneities within HP1 condensates. The first half would benefit from mechanistic understanding of how the PEG result relates to other FLIM mCherry experiments in the field e.g. does PEG alter homo-FRET and the second half would benefit from supporting evidence that HP1 density does change. It is at present not clear that the PEG effects in the first half are directly linked to the mCherry lifetime changes in the second half. Maybe the correlation of density with lifetime in live cells would help?

Minor comments:

TYPO pg 4 paragraph before Results: authors want to say shift to "single-phase" liquid-like condensates

Reviewer #3 (Remarks to the Author):

The paper describes a potential use of some monomeric proteins (mCherry, mRFP) as sensors of local crowding, through the reduction of their fluorescence lifetime. If true, this could be an exciting discovery, providing the community with encodable ways of detecting crowding in liquid liquid phase separated (LLPS) coacervates in biology, e.g. in nuclear condensates, that were attempted to be visualised in the present study.

I am not entirely persuaded by the evidence presented for the following reasons:

- 1) The photophysical behaviour of the proteins doesn't quite tally. While mCherry and mRFP are insensitive to viscosity and sensitive to crowding, eGFP and mCitrine are sensitive to the refractive index and insensitive to crowding. I am puzzled by this observation, particularly as the GFP chromophore is known to be very sensitive to the degree of confinement and would be expected to respond strongly to the change in the cavity shape/size. Why didn't the authors perform molecular dynamics/modelling studies for all the proteins in question, to demonstrate that mCherry and mRFP are unique in their cavity response? Only mCherry data is presented currently
- 2) Following on with the photophysics, while eGFP and mCitrine show a strong dependence of the lifetime on the refractive index, as expected from the Strickler Berg formula, the MCherry and mRFP do not show such dependence. Why is this the case? The authors quote 'most probably due to their lower structural stability relative to the red fluorescent proteins'. I do not understand this explanation – some other factors must be at play, as the dependence of the rate constant on the refractive index square is universal. Have the authors considered all possible radiative and non radiative decay channels? This is important, since LLPS condensates are likely to be characterised by very high refractive indexes and it is vital to understand why the very well understood physics of the Strickler Berg formula does not work on the present probes (or, if it does, what other factors are superimposing)
- 3) Have the authors tested the effect of varying concentration of mCherry? Higher concentration quenching and consequential reduction in lifetime are often observed, and increased crowding can be

exacerbating some of these effects. Furthermore, the concentration of the fused protein in the nucleus would be poorly controlled, potentially causing heterogeneity observed in FLIM, which was interpreted as a variety of crowding environments for mCherry. At the same time, the same concentration variation is not likely to affect the bleached FCS data, taken as a proof of diffusivity of the constructs.

4) I note that the change in lifetime is relatively minor for all but the highest crowding conditions (50% FVO) for both 'responsive' proteins, and unseen in cells. There is a further ambiguity with the fact that sometimes the lifetime changes upwards, eg in the presence of glycerol. This makes me think that other factors (polarity, dielectric?) might be at play.

In conclusion, the observations reported here are interesting, and complemented by an attempted study of interesting biology that involves LLPS of chromatin. However, I feel that the characterisation of the probe is not complete and hence it is not possible to confidently assign the observed features to the crowding phenomena.

REVIEWER COMMENTS

We would like to thank the reviewers for their assistance in the review of our work, which undoubtedly improved it. All text changes are yellow-highlighted in the resubmitted main text, as well as shown in this response letter.

Reviewer #1 (Remarks to the Author):

The authors present an excellently written and comprehensive study of the application of fluorescent proteins as sensors of phase transitions in the nucleus. They present fluorescence lifetime measurements as a valuable tool to study phase separations/transitions and changes in molecular crowding during chromatin activation in the differentiation process of ESC. In the first part, the authors perform all the necessary control studies to characterize and differentiate the effects of refractive index and molecular crowding on the fluorescence lifetime of fluorescent proteins. Interestingly, mCherry exhibits a different behavior than mCitrine and EGFP.

In the second part, experimental data and molecular modeling are used to convincingly demonstrate how a change in molecular density in the environment can contribute to a faster de-excitation of the excited fluorophore.

Eventually, in the third part, FLIM was employed to determine differences in heterochromatin foci in the nucleus of undifferentiated and retinal-stimulated ESC. The authors found differences in the fluorescent lifetime of mCherry-HP1 α distribution, with a less heterogeneous distribution of tau in differentiated cells. They suggest that HP1 is found in foci that are characterized by different subdomains within the foci and distinct molecular dynamics.

The study is well-designed and conclusive.

The demonstration and professional introduction of the FLIM technique allow its application beyond the specific study and is of general interest since it advertises FLIM as a method for imaging and identifying phase transitions in living cells.

We suggest publication after a minor revision.

We would like to thank the reviewer for his/her kind words and summary of our work.

Minor:

Please comment on the time course of the anisotropy decrease at high PEG (Fig. 2b). Do you expect a further increase in anisotropy after 20 μ s? This would correlate with your proposed greater freedom.

We apologize for not making it clear enough, and based on the reviewer's comment, we have modified the text to better explain this point. The anisotropy decay is calculated as a ratio of differences and sums of fluorescence decays (parallelly- and perpendicularly-polarized fluorescence decays). Each fluorescence decay has its own experimental noise. Therefore, the ratio of fluorescence decays is even noisier. The 10 ns window shown for the fluorescence anisotropy decay was the maximal time window that could still show the anisotropy decay trend with bearable noise. The increased rapid rotational degrees of freedom of the fluorophore are relative to the beta-barrel fold, on top of the slow rotational mode of the tumbling of the beta-barrel fold in space. In the manuscript, we report that as long as the PEG %FVO is below the 30% threshold, the only changes we observe are due to the slow tumbling of the protein becoming slower with the increase in viscosity. This is expected due to the increased viscosity that PEG introduces, which decreases the rotational diffusion following the Stokes-Einstein relationship. However, no rapid rotational component is observed; only above 30% FVO.

Accordingly, under the subtitle "Mechanism of fluorescence lifetime reduction in FPs" (p. 7) of the revised manuscript we added the following text (additions are in **bold**):

We performed fluorescence anisotropy decay measurements of mCherry (Fig. 2a-b) in 1) 0% FVO PEG 6,000 (buffer only), 2) in 25% FVO PEG 6,000, **just slightly below the 30% FVO limit**, where only the increased **bulk** viscosity is expected to slow down motions and fluorescence lifetimes do not yet change (Fig. 1e), and 3) in 50% FVO PEG 6,000, where bulk viscosity is further increased, and also fluorescence lifetime decreases (Fig. 1e), yet potentially also the internal degrees of freedom of the fluorophore relative to the β -barrel fold of the FP (Fig. 2a, red, Fig. 1b). The fluorescence anisotropy decay was then calculated from the fluorescence decays **polarized parallel and perpendicular to the excitation light plane of polarization** (Supplementary equation (3)). **The 10 ns window shown for the fluorescence anisotropy decay is the maximal time window that could still show the decay trend with bearable noise (Fig. 2b)**. Since the fluorophore system is constricted inside the FP, it is expected that mCherry in buffer will exhibit a fluorescence anisotropy decay with mostly a slow decaying component representing the slow tumbling of the whole FP, **which is further slowed by the increase in viscosity due to the increase in PEG concentration** (Fig. 2b, black).

Fig. 4: Please clarify if the precipitate environment is still wet or if it has dried out.

We thank the reviewer for this comment. The environment was wet. We now added “*in solution*” at all relevant instances, to clarify that we are not dealing with the unique case of a dried-out precipitate.

Page 9, last paragraph before new subtitle (fluorescence lifetimes lower than their standard values can report on local densities or crowdedness per pixel of an image), please include measured tau values in the text showing the difference described.

The measured tau values were added to the text, as requested.

Mandatory: the authors mention a study (Novo et al.) showing that heterochromatin condensates have the characteristics of homogeneous condensates after depletion of MSR transcripts. Determining the tau value of the probe under these conditions would be a valuable contribution and would support your interpretation of the data.

We thank the reviewer for this comment, which prompted us to attempt to perform the MSR transcript depletion experiments in ESCs and acquire FLIM images. Reassuringly, adding the LNA-based probes from Novo *et al.* to undifferentiated ESCs resulted in FLIM images with similar phenotypes as those after exposing ESCs to RA. Scrambled RNA control probes, which served as controls, did not induce any change. These results have been added to the main text (in the introduction, in the results, and in the discussion), Figures, and the SI.

The following are examples to text we added to describe these experiments:

1. At the end of the introduction (end of page 4 in the revised manuscript):
However, HP1 α condensates in retinoic acid (RA)-treated ESCs or in ESCs in which major satellite repeats (MSR) were depleted, exhibit lower and less heterogeneous densities, suggesting a shift towards a single liquid-like phase.
2. As an additional paragraph at the Results' section “The phase of HP1 α condensates in undifferentiated and early differentiated ESCs” (pages 12-13 in the revised manuscript):

In mouse ESCs, MSRs are highly-transcribed^{66,67} and play a major role in defining heterochromatin structure and its regulation²⁹. After transcription, MSRs remain in close proximity to chromocenters and have been shown to interact with heterochromatin-associated proteins, such as HP1 α , and promote its phase separation²⁹. Therefore, by transfecting undifferentiating ESCs with locked nucleic acid (LNA) oligonucleotide probes that deplete MSR transcripts with an efficiency of ~50% (gampers)^{29,68}, thereby affecting their interactions with chromocenters, we expect mCherry-HP1 α condensates to exhibit results closer to those observed after RA-induced early-differentiation, hence less heterogeneity in fluorescence lifetimes and higher fluorescence lifetime values. Reassuringly, undifferentiated ESCs treated with the MSR gampers exhibited increased pixel-wise fluorescence lifetime values, similar to

the typical mCherry lifetime values below 30% FVO (i.e., ~1.44-1.50 ns), and less heterogeneity when compared with non-targeting scrambled RNA control probes (Fig. 7a-l). HP1 α condensates were less dense with lower density heterogeneity, very similar to the results in RA-induced early-differentiated ESCs (Fig. 5g-l). Performing the same transfection of the MSR-interacting LNA probes on RA-induced early-differentiated ESCs, revealed no significant effect on condensate formation and the low heterogeneity was maintained (Fig. 7m-r). We also examined the mean fluorescence lifetimes over all pixels of a condensate and compared these values between multiple HP1 α condensates in multiple image acquisitions of undifferentiated ESCs and early-differentiating ESCs transfected with the MSR gapmers. ESCs treated with the gapmers showed increased mean fluorescence lifetime and reduced heterogeneity of lifetimes, similar to the results after RA induction. As a negative control, we performed similar analyses on ESCs transfected with the scrambled RNA probes and observed no change in the mean lifetimes or their heterogeneities between condensates (Supplementary Fig. 17), relative to the results in the absence of any transfection.

3. In the discussion (page 17 in the revised manuscript):

Recently, Novo *et al.* demonstrated that heterochromatin condensates, comprising MSR transcripts and HP1 α , form in undifferentiated mouse ESCs²⁹. Interestingly, these condensates exhibited features that can be partially explained as induced by LLPS towards an interior single liquid phase and others that cannot (e.g., irregular shapes, substantial immobile fraction), and hence coincide with the heterogeneities we report here. Interestingly, upon MSR transcript depletion, the heterochromatin condensates exhibited characteristics of homogeneous condensates. We have previously shown that ESC differentiation leads to the reduction in such noncoding satellite repeat transcripts⁹¹, and hence the Novo *et al.* findings²⁹ coincide also with our own regarding early differentiated mouse ESCs. To test this directly, we employed the same LNA gapmer probes used in the Novo *et al.* study to suppress MSR expression in ESCs. Since MSR transcripts promote HP1 α phase separation²⁹, their suppression should eliminate at least one of the phase-separated HP1 α populations in the heterogeneous undifferentiated state. Supporting this view, we observed a transition from a multi-phase state to a liquid-like state similar to the situation in early differentiating cells.

4. In the acknowledgments (page 25 in the revised manuscript): We thank Genevieve Almouzni for kindly providing the gapmers for MSRs

Importantly, these results are also summarized in new figure 7 and in supplementary figures 16 & 17.

Reviewer #2 (Remarks to the Author):

The authors use fluorophore lifetime as an assay to address changes in density within biological systems. The authors begin by showing in vitro that the fluorescence lifetime of mCherry and other fluorophores are decreased by molecules that increase molecular crowding. They attribute this to an increase in the degrees of freedom of the fluorophore within the fluorescent protein based on anisotropy measurements and molecular simulations. They then use

this to show changes in the densities of HP1alpha condensates in undifferentiated and early differentiated ESCs.

We would like to thank the reviewer for his/her kind words and summary of our work.

Main comments:

Overall, the use of FLIM to study heterogeneities within condensates is exciting. FLIM is a sensitive tool for monitoring regions within live cells and so it would indeed be nice to use it to report on heterogeneities in density with condensates, especially since one could combine this with other imaging modalities e.g. to show that RNA within heterochromatin condensates drive these inhomogeneities. The biological conclusion is also exciting in principle with the idea that HP1 condensates go from heterogeneous condensates to a homogeneous state as ESCs start to differentiate. A recent study they reference highlighted changes in RNA within ESC HP1alpha condensates that could be driving these differences.

We would like to thank the reviewer for finding the use of FLIM to study heterogeneities within condensates exciting and for thoroughly examining our manuscript.

Sadly, enthusiasm for the paper was somewhat reduced by some of the issues described below:

1. I found the intro confusing as it started with the basics of a fluorophore before we realise the authors are looking for a way to use fluorescent lifetimes as a density readout. It may be better to describe the alternate approaches for studying density, then discuss how FLIM is being used to address this and then discuss fluorophores and their lifetime, introducing papers using this approach and explaining how this paper contributes to the field.

We would like to thank the reviewer for bringing up this important point. We have now modified the introduction to present both aspects of this work (FP aspect and PS condensate aspect) in an orderly manner and cite the relevant literature. Importantly, throughout this revision, we added more experimental results, which required expanding the last paragraph of the introduction a bit further.

2. The authors use molecular simulations to show that PEG is altering protein conformation (Fig 3) and that this is consistent with anisotropy changes (Fig 2) and lifetime (Fig 1). These experiments are very clear and well done. I also like the discussion making clear that some of the other fluorophores in Supplementary are affected by glycerol too so there may be other mechanisms

at play for these fluorophores. The idea of using fluorophore lifetime or anisotropy to study the density of fluorophore-tagged proteins is not novel per se (PMID: 26133241; PMID: 35298090; PMID: 24703307). There have even been attempts to improve mCherry quantum yield through mutations that affect lifetime (PMID: 35709514). Having said that, previous studies have focussed on how FRET or homo-FRET affects lifetime/anisotropy and taken advantage of these changes to readout local density i.e. homo-FRET reduces fluorophore lifetime and affects anisotropy measurements. PMID: 26133241, for example, uses changes in the lifetime of mCherry, one of the proteins studied here, but attributes the change in lifetime to homo-FRET. It seems that the contribution of this paper is to show that PEG and crowding can itself affect FP conformation and therefore fluorophore anisotropy. It may therefore be important for the authors to explain how their results relate to previously proposed homoFRET based changes. Does PEG affect fluorophore intensity/quantum yield or alter FRET efficiency?

We thank the reviewer for pointing out the homoFRET and anisotropy work on sensing density/crowdedness. We are aware of this possibility but did not discuss this point in the original manuscript, because we found that the fluorescence lifetimes of FPs solely are enough to report on densities or crowdedness levels. Nevertheless, we now cite these works.

In the simplest terms, fluorescence anisotropy measures how well a fluorophore can depolarize its emitted light relative to the orientation of polarization of the excitation light, which photoselects the molecules to be excited at a given moment (the moment of excitation). According to the Perrin equation (see newly added supplementary equation 4), the value of the fluorescence anisotropy, r , is linearly proportional to the rotational correlation time, θ , or how fast its depolarization occurs, and inversely proportional to the fluorescence lifetime, τ . Therefore, factors that influence θ are the typical factors in the Stokes-Einstein equation (temperature, viscosity, size/volume) but also factors such as homoFRET. In homoFRET, the energy transfer can occur between donors and acceptors with their dipoles not necessarily parallelly oriented. Therefore, excitation energy from one dipole (that was aligned with the polarization of the excitation light at the moment of excitation) to another dipole (with a dipole not necessarily aligned with the polarization of the excitation light) can lead to faster depolarization of light. The main observable in homoFRET are reductions of fluorescence anisotropy, r , due to reductions in the rotational correlation time, θ , and not due to increases in the fluorescence lifetime, τ . Fluorescence intensities and lifetimes are, by default, not observables of homoFRET - when the donor dye "loses" fluorescence, the lost

fluorescence is expressed in “additional” fluorescence of the acceptor, but both donors and acceptors share the same fluorescence spectra, and so intensities and lifetimes stay the same. Of course, homoFRET, as well as heteroFRET, could aid in reporting high local densities within PS condensates, since FRET occurs effectively only if a high enough amount of FPs-tagged proteins are <10 nm from each other.

We are thankful to the reviewer for bringing the work from Warren *et al.* to our attention since the authors of this work perform homoFRET experiments on mCherry-tagged proteins and report changes both in anisotropy values and lifetime values. While the former is a clear readout of homoFRET, the latter is unexpected. The findings were indeed attributed to changes in densities. However, going over this work, we did not find efforts to explain the underlying mechanism for the fluorescence lifetime changes due to differences in densities.

Most importantly, in the revised manuscript we have added anisotropy images of undifferentiated and early-differentiated ESCs, and the heterogeneity of fluorescence anisotropy values with the HP1alpha foci as well (see new supplementary figure 15). These results show significantly reduced anisotropy values in undifferentiated ESCs relative to the values in early-differentiating ESCs, which could suggest denser organizations in undifferentiated ESCs due to more homoFRET. However, this might be true in the case where no lifetime variations would have also been reported, unlike in our case. We explain this in the text - that while anisotropy exhibits heterogeneities, and they arise most probably from the heterogeneity in densities, we cannot tell whether they are caused by heterogeneities in homoFRET between mCherry proteins or due to changes to fluorescence lifetimes, and explain that homoFRET is sensitive to <10 nm proximities, while lifetime changes are probably sensitive to shorter proximities.

In summary, we thank the reviewer for this insightful and thorough examination. In the revised text we:

1. Include citations of previous FRET imaging (homoFRET & heteroFRET) works that assisted in reporting local densities, and explain their pros and cons
2. Explain what is homoFRET, what is the main readout, and why the finding on mCherry lifetime changes due to changes in densities, without FRET alone, are novel, and most importantly, cite Warren et al. for their finding about FP lifetime changes, which was suggested to be related to density changes

3. Provide anisotropy images that recover density heterogeneities observed also in FLIM, to show that there are two factors that could influence fluorescence anisotropy values (lifetimes and homoFRET), and it is unclear how to decouple them. While homoFRET reports on nanocompaction, hence on compaction leading to proximities <10nm, the mechanism we find for lifetime reductions due to increased densities is also due to condensate nanocompaction, but perhaps at the shorter proximities that lead to contacts between proteins. While we cannot decouple homoFRET from lifetime variations, FLIM images in an experiment with one type of FP in condensates report directly on high densities in condensate nanocompaction due to one mechanism.

Accordingly, we have added the following passages, accordingly:

1. In the results' section "Mechanism of fluorescence lifetime reduction in FPs" (page 7 in the revised manuscript):

Moreover, according to the Perrin equation (Supplementary equation (4)), if there is a single mode of rotation, a decrease in fluorescence lifetimes should introduce an increase in fluorescence anisotropy, which is different from the observation. In fact, following the Perrin equation, the observed faster decay of anisotropy can be explained by a fast-decaying rotational correlation component.

2. In the results' section "The phase of HP1 α condensates in undifferentiated and early differentiated ESCs" (pages 11-12 in the revised manuscript):

Importantly, since the elevated densities inside PS condensates are expected to be high enough to bring mCherry-HP1 α constructs within 10 nm proximities, in which homoFRET could occur, we also performed fluorescence anisotropy imaging. Our results show that in undifferentiated ESCs the fluorescence anisotropy values within mCherry-HP1 α condensates are lower than in early-differentiated ESCs (Supplementary Fig. 15). In situations where no fluorescence lifetime changes are observed, these results would indicate higher degree of homoFRET in undifferentiated ESCs than in early-differentiated cells, serving as yet another indication of higher condensate densities in undifferentiated ESCs. Nevertheless, since we do observe fluorescence lifetime variations (Fig. 5), according to the Perrin equation (Supplementary equation (4)), homoFRET cannot be easily decoupled from fluorescence lifetime effects on the fluorescence anisotropy values. However, the magnitude of the difference in fluorescence anisotropy values between undifferentiated and early-differentiated ESCs is larger than expected from variability in fluorescence lifetime variations solely. Therefore, more homoFRET, hence denser condensates, occur in undifferentiated ESCs than after early differentiation. Interestingly, fluorescence anisotropy values in undifferentiated ESCs are more heterogeneous than in RA-induced early-differentiated ESCs, suggesting that the observed results are merely mirroring the fluorescence lifetime images shown above (Fig. 5).

3. In the discussion (page 14 in the revised manuscript):

Another FRET-based approach to sense nanocompaction and high densities is the use of homoFRET, which relies on the transfer of excitation energy between two same fluorophores, owing to their proximity within 10 nm and the overlap between their excitation and emission spectra. While in bio-condensates, it is plausible to expect FRET-relevant distances between nearby FP fluorophores, in homoFRET upon energy transfer, donor fluorescence decreases and in parallel, acceptor fluorescence increases. However, the same spectral region is covered for both donor and acceptor emission, which disables the use of homoFRET to be reported via changes in fluorescence intensities or lifetimes. Still, homoFRET can be reported via reduced fluorescence anisotropy values, since energy transfer does occur between same type fluorophores with different orientations in space, which further contributes to the depolarization of fluorescence, and hence to the reduced fluorescence anisotropy values. Yet, if fluorescence lifetimes of the fluorophores do not change, the sole reason for reductions in fluorescence anisotropy values would be the reduction in the rotational correlation time due to faster depolarization of fluorescence, as can also be understood from the Perrin equation (Supplementary equation (4)). Still, in our case, we observe also fluorescence lifetime variations, and therefore cannot report solely on changes of homoFRET in fluorescence anisotropy imaging. Interestingly, previous reports of sensing inner densities using fluorescence anisotropies have reported both changes in fluorescence anisotropies and lifetimes of FPs, and reported them to be caused by homoFRET, where the more homoFRET occurs, the more dense the FP-fused proteins are³³. However, the fluorescence lifetime changes reported in that work could not be explained by a homoFRET mechanism, and were not explained mechanistically. We suggest that this previous study of inner densities observed FLIM results similar to ours but did not provide a mechanism to explain the FP lifetime variations dependence on high inner densities. Accordingly, we propose using FLIM to image proteins genetically fused to typical monomeric FPs that exhibit fluorescence lifetime reductions mostly in response to increased density and crowding above a given FVO threshold, to report on local densities inside PS condensates.

3. Next, they use Fig1-3 results as a justification for FLIM in Fig 4-6. However, it is not clear if the same mechanism of crowding is affecting the lifetime measurements inside living cells. Possible experiments (not all are required) that could make this more convincing include correlations between HP1 condensate intensity and lifetime, live-cell measurements of anisotropy or orthogonal evidence that HP1 is denser in ES cells (does the intensity within HP1 foci relate to the lower lifetime measurements?).

We would like to thank the reviewer for pointing out this lack of clarity in the results. We believe the new figures added following additional experiments showing only a weak correlation at most between fluorescence intensities and lifetimes within HP1 condensates will help clarify how the suggested mechanism of crowding that's affecting the lifetime inside living cells (see new supplementary figure 14). This means that out of the two major contributions to fluorescence intensity (i.e., amount of fluorescently-active FP and fluorescence lifetimes), the fluorescence intensities are mostly due to the differences in amounts of fluorescently-active FPs within PS condensates, and the lifetimes report on the fine differences in close proximity between nearby FPs. To explain this, we have added the following passage to the results' chapter

“The phase of HP1 α condensates in undifferentiated and early differentiated ESCs” (page 11 in the revised manuscript):

It is noteworthy that pixel-wise fluorescence lifetime values are at most weakly correlated with pixel-wise fluorescence intensity values (Supplementary Fig. 14).

As well as in the discussion (page 14 in the revised manuscript):

It is noteworthy, that apart from fluorescence intensity scaling as a function of the number of FPs within an imaged pixel, it is also influenced by fluorescence quantum yields and lifetimes. Nevertheless, at least in our results, pixel-wise fluorescence intensity and lifetime values do not exhibit a dramatic correlation (Supplementary Fig. 14).

Additionally, we added anisotropy images of HP1 α PS condensates in undifferentiated and early-differentiated ESCs (see new supplementary figure 15), exhibiting: (i) more homoFRET in undifferentiated ESCs than in early-differentiated ESCs, implicating density changes, and (ii) more heterogeneities of fluorescence anisotropy values in undifferentiated ESCs than in early-differentiated ESCs, which could suggest changes in density heterogeneities. However, we discuss why it is difficult to use fluorescence anisotropy to decouple the different sources (e.g., levels of homoFRET, fluorescence lifetimes) from the heterogeneities in anisotropy. Fluorescence anisotropy images of mCherry-tagged HP1 foci in ESC cells after exposure to retinoic acid, in fact, exhibit anisotropy values relative to those in undifferentiated ESCs, as well as reduced anisotropy heterogeneity, the same way lifetime heterogeneities reduce in the FLIM images. The anisotropy values still suggest homoFRET occurs within mCherry-HP1 condensates in undifferentiated ESCs (for a summary of the changes related to this point, see the response given above).

Lastly, in the revised manuscript, we added results from experiments using MSR transcript depletion probes, which were already reported by Novo *et al.* to induce the transition in HP1 α PS condensates similar to that achieved after RA-induced differentiation of ESCs. The results (summarized in the text and in new figure 7 and new supplementary figures 16 & 17) show that using FLIM we recover similar observables with MSR depletion relative to undifferentiated ESCs alone, as we recover for early-differentiated ESCs (for a summary of the changes related to this point, see the response to reviewer #1).

4. Finally, the authors conclude from Figure 5-6 that HP1 condensates are heterogeneous in ESCs prior to differentiation. Although exciting if true, the authors show data from condensates within 5-15 cells. It is unlikely that this accounts for cell cycle which is known to affect condensate formation so at least 20 cells would be needed to be sure of this result. More cells may allow multi-Gaussian fitting and BIC analysis to demonstrate that there is more than one

peak to the undifferentiated inside foci. It should be noted in the discussion that cell cycle will change during differentiation which could also affect the results.

We thank the reviewer for this important suggestion. This suggestion urged us to (1) have a deeper look at our previous analyses, (2) reanalyze the existing imaging data, and (3) analyze data from additional new acquisitions. We now report on the differences between ESCs before and after RA-induced differentiation from more cells and more foci (173 foci from 137 undifferentiated ESCs; 271 foci from 133 early-differentiated ESCs; see new figure 6). Importantly the results of our reanalyses are more meaningful statistically and the differences are now even sharper and more significant (we report the p-values of the differences between histograms in means and variances). We have also added another control, following the suggestion of reviewer #1, which allows us to test the fluorescence lifetimes after MSR transcript depletion, which was previously reported to induce conditions that yield similar behavior as after RA-induced differentiation, and indeed exhibit similar values shown after induction with RA (90 foci from 23 ESCs exposed to MSR transcription depletion LNA probes; see new supplementary figure 16, as well as new figure 7 for examples of such acquisitions).

Importantly, we observed lifetime differences between the previous analyses and the current reanalyses, which shifted to slightly higher values, owing to more rigorous reanalyses of pixel-wise fluorescence lifetimes.

We explain all about these modifications in the text (see responses above).

5. Finally, there is a feeling that there are two papers in one here and the authors should make an attempt to better connect the two halves. The first half delves into the mechanism of lifetime changes when mCherry density is increased and then the second half is about using FLIM to show heterogeneities within HP1 condensates. The first half would benefit from mechanistic understanding of how the PEG result relates to other FLIM mCherry experiments in the field e.g. does PEG alter homo-FRET and the second half would benefit from supporting evidence that HP1 density does change. It is at present not clear that the PEG effects in the first half are directly linked to the mCherry lifetime changes in the second half. Maybe the correlation of density with lifetime in live cells would help?

We would like to thank the reviewer for raising this point. As mentioned above, this is because the manuscript presents two main findings, one which is a major methodological message regarding FLIM of monomeric FPs, and the other, which is an application of the methodological development for quantifying

density homo/heterogeneities within condensates. This order is by design. However, in the revised text we made efforts to better link the two stories/findings.

Minor comments:

TYPO pg 4 paragraph before Results: authors want to say shift to "single-phase" liquid-like condensates

Done.

Reviewer #3 (Remarks to the Author):

The paper describes a potential use of some monomeric proteins (mCherry, mRFP) as sensors of local crowding, through the reduction of their fluorescence lifetime. If true, this could be an exciting discovery, providing the community with encodable ways of detecting crowding in liquid-liquid phase separated (LLPS) coacervates in biology, e.g. in nuclear condensates, that were attempted to be visualised in the present study.

We thank the reviewer for finding our method to be potentially exciting. In the revised manuscript, we added more data, figures, and detailed explanations, and hope the reviewer will find our work suitable for publication. Please see below (and above) for detailed responses.

I am not entirely persuaded by the evidence presented for the following reasons:

1) The photophysical behaviour of the proteins doesn't quite tally. While mCherry and mRFP are insensitive to viscosity and sensitive to crowding, eGFP and mCitrine are sensitive to the refractive index and insensitive to crowding. I am puzzled by this observation, particularly as the GFP chromophore is known to be very sensitive to the degree of confinement and would be expected to respond strongly to the change in the cavity shape/size. Why didn't the authors perform molecular dynamics/modelling studies for all the proteins in question, to demonstrate that mCherry and mRFP are unique in their cavity response? Only mCherry data is presented currently

We would like to thank the reviewer for raising these concerns. We believe the message about our findings was perhaps not fully communicated. It is not that viscosity/refractive index/crowding effects do or do not affect the fluorescence lifetimes of the different FPs tested. The question is more about the magnitude of these effects on the different FPs that were tested. We break down our answer to the different parameters tested:

- The Strickler-Berg equation discusses the potential effect of changes to the refractive index in the vicinity of the fluorophore on the fluorescence lifetime, not necessarily changes to the bulk refractive index. This is because the Strickler-Berg relationship refers to the dependence of fluorescence lifetime on the local refractive index (i.e., in the vicinity of the fluorophore). The reviewer is correct that small changes to the microenvironment of the fluorophore inside the FP can lead to large changes in fluorescence and one of these changes will be driven by changes to the local refractive index, which will then influence the rate of radiative de-excitation following the Strickler-Berg relationship. Therefore, the titrations of different conditions that we tested against fluorescence lifetimes of FPs, which introduce increases to the bulk refractive index, still might not introduce changes to the local refractive index of the fluorophore. It is possible that while we increase the concentration of additives and introduce increases to the bulk refractive index, especially in the case of Glycerol, PEG, and Trehalose titrations, these might not influence the local refractive index of the fluorophores that are buried within the FP, which then might not influence the fluorescence lifetime values through changes in the radiative de-excitation rate. If there was an effect on the local refractive index of the fluorophore within the FP, we would have expected from the Strickler-Berg relationship to observe a monotonic decrease of the fluorescence lifetime with the increase in the additive concentration, already from the initial additive concentrations, that would follow the trend of the increase in bulk refractive index. Indeed, for eGFP, and, to a lesser extent for mCitrine, we see such monotonic decrease trends in fluorescence lifetime values as Glycerol and PEG are introduced, even at low concentrations (see supplementary figures 5 & 6). However, for mRFP and mCherry such monotonic trends are not observed. For Glycerol and PEG these monotonic lifetime decrease trends are observed only above a given %FVO value, for which the bulk refractive index is already very high (and for PEG, the bulk refractive index is almost unchanging starting from 5% FVO). To explain these points we also provided references to the known changes in the bulk refractive index as a function of additive concentration, which is different than the lack of change in fluorescence lifetime we observe in mRFP and in mCherry. Following, we concluded that in mCherry & mRFP the local refractive index of the fluorophore is potentially largely unaffected as the bulk refractive index does, at least below 30% FVO, and that this is true to a lesser extent in mCitrine and to an even lesser extent to eGFP.

- For these reasons, we can conclude that
 - The FPs that were tested exhibit a reduction in fluorescence lifetimes due to crowdedness above a given %FVO, aside from other factors (e.g., viscosity, refractive index) that influence some FPs (mostly in eGFP and mCitrine)
 - In mCherry & mRFP, crowdedness is the major driver of the reduction in fluorescence lifetime and hence could be used to map back crowdedness levels in its vicinity

Therefore, in the revised text, we have rewritten these passages for the sake of clarification of these explanations:

1. In the results' chapter "Fluorescence lifetime reductions in FPs due to elevated crowdedness levels" (changed text is presented in **bold**; pages 5-6 in the revised manuscript):

Importantly, increasing the concentration of kosmotropes and macromolecular crowding agents also increase the refractive index in the bulk solution. If the refractive index in the vicinity of the fluorophore inside the β -barrel also increases, that could also introduce reduction in the fluorescence lifetime, which can be explained by the Strickler-Berg relationship³¹. Interestingly, this effect did not clearly show up in fluorescence lifetimes of mCherry (Fig. 1b-f), even though the bulk refractive index increases at the tested conditions. **We note that this can happen if the local refractive index inside mCherry, in the vicinity to its fluorophore, is not influenced from the bulk refractive index.**

2. In the results' chapter "Fluorescence lifetime reductions in FPs due to elevated crowdedness levels" (changed text is presented in **bold**; page 6 in the revised manuscript):

Encouraged by the results of mCherry, we performed similar analyses for other common monomeric FPs (e.g., mRFP, mCitrine, eGFP), which exhibited similar trends as mCherry, relative to their standard fluorescence lifetime values (Supplementary Figs. 4-6), albeit with different value ranges. Interestingly, eGFP and mCitrine reported fluorescence lifetime reductions starting from values of ~ 2.6 - 2.7 ns^{44,42,45} and ~ 3.4 - 3.6 ns⁴⁶⁻⁴⁹, respectively, also as a function of viscosity (Supplementary Figs. 5-6), **unlike in mCherry and mRFP**. Yet, reductions of fluorescence lifetimes of eGFP and mCitrine below the lifetime values at 50% glycerol still occur in the presence of PEG at crowding levels higher than 30% FVO (Supplementary Figs. 5-6). Could this be **due to changes in the local refractive index in the vicinity of the fluorophore groups inside mCitrine and eGFP?** For PEG 6,000 and similar PEGs, the **bulk** refractive index reaches an asymptotically constant value of ~ 1.46 at an FVO as low as $\sim 5\%$ ⁵⁰. Therefore, it is clear that the fluorescence lifetime reductions in eGFP and mCitrine due to crowding **occur on top of reductions potentially due to changes to the local refractive index**. Indeed, it has been reported that increase of the **bulk** refractive index leads to reductions in the fluorescence lifetimes of eGFP⁵¹ and mCitrine⁵². All of the physicochemical conditions that were tested induce significant enhancements of the bulk refractive index, including molar concentrations of NaCl. However, such high NaCl concentrations did not induce reductions in monomeric FPs fluorescence lifetimes (Fig.1, Supplementary Figs. 4-6). Nevertheless, even the high NaCl concentrations used here do not

introduce increases in bulk refractive index⁵³ as high as those introduced by PEG⁵⁰. In addition, the lifetime trends observed as a function of PEG are occurring while the bulk refractive index is expected to stay almost unchanged (**i.e., PEG above 5% FVO**)⁵⁰. We can therefore conclude that the major reductions in fluorescence lifetimes above a given concentration **or FVO** threshold can be attributed to crowding. Importantly, we can conclude that in mCherry and mRFP, fluorescence lifetime reductions cannot be attributed to changes in viscosity or local refractive index, but rather to crowding alone.

3. In the discussion (changed text is presented in **bold**; page 15 in the revised manuscript):

Unlike mCherry and mRFP, our **tests** show that the fluorescence lifetimes of eGFP and mCitrine may be influenced not **predominantly** by crowding but also by an increase in the **local** refractive index near the fluorophore inside the **β -barrel fold**. This effect is established in the Strickler-Berg relationship between the rate of radiative de-excitation of the fluorophore and the square of the refractive index in the vicinity of the fluorophore, and hence depends inversely on the fluorescence lifetime. We have shown that this effect **is observable** in green-yellow FPs (e.g., eGFP, mCitrine; Supplementary Figs. 5, 6). **We propose that this is mostly attributed to the previously reported lower structural stabilities of green-yellow FPs relative to red FPs (e.g., mRFP, mCherry), where this effect was negligible⁷⁵. In short, we propose that the higher the structural stability of the β -barrel fold, the less the local refractive index in the microenvironment of the fluorophore will be influenced by changes in the bulk refractive index, which would then lower the impact of the Strickler-Berg effect on the radiative de-excitation pathway.** Therefore, we propose to use fusions of mCherry, mRFP, or other monomeric FPs that exhibit fluorescence lifetime reductions mostly or solely due to crowding. Additional monomeric FPs designed to be highly stable have been reported in the literature⁷⁰ and are potential subjects for further investigations similar to those presented here.

Importantly, this is the reason why our further examinations focused solely on mCherry and not on the other FPs that were experimentally tested. Most of our live cell images were also performed using mCherry, which comes to explain why after surveying the different FPs, we focused on mCherry. We absolutely agree with the reviewer that the modeling and further examinations of the other FPs, as well as additional FPs, we did not test in this work, will serve as important tests in follow-up works. However, the brevity of the manuscript, in addition to our efforts not to defocus this work beyond its current status make it suitable to perform such modeling tests in our next works. To express this, we added the following passages in the discussion:

1. **Page 16 in the revised manuscript:** Importantly, while we got closer to a working mechanism of how mCherry might act as a fluorescence lifetime sensor of high density, the mechanism of how other monomeric FPs might act as density sensors will be the subject of a follow-up study, inspired by the results of this work.
2. **Pages 17-18 in the revised manuscript:** While we report a methodological concept, provide a working mechanism for it, and demonstrate its use, many questions are yet to be answered. On the FP side, how many other FPs can also exhibit the effects shown here by mCherry and mRFP? Could the reported crowding effects on internal

fluorophore degrees of freedom be mutationally designed? On the application of FPs to study PS condensates, what further phase transitions could be studied with this approach combined with other experimental approaches? and most importantly, could the herein presented experimental approach assist in achieving a clear picture of PS condensates in cells? We envision these questions as well as others would be addressed in future studies building on this work.

Furthermore, the simulations of mCherry carried out in this study involved more than three times as much sampling as the most recent simulation study of mCherry and mCherry variants (see <https://doi.org/10.1021/acs.jpcb.2c05584>), and repeat these simulations for multiple FPs is not possible with currently available computational resources. Importantly, the clear correlation between our findings and previous characterizations of FP thermodynamic stabilities could serve as a major explanation for the differences we observe (which we discuss in the discussion), albeit perhaps not the only explanation. One of the reasons we actually think the red FPs might serve as better indicators is exactly because they are more thermodynamically stable, and hence relying on changes in their fluorescence characteristics should in principle be more telling than those in eGFP (which our survey continues to prove).

2) Following on with the photophysics, while eGFP and mCitrine show a strong dependence of the lifetime on the refractive index, as expected from the Strickler Berg formula, the MCherry and mRFP do not show such dependence. Why is this the case? The authors quote ‘most probably due to their lower structural stability relative to the red fluorescent proteins’. I do not understand this explanation – some other factors must be at play, as the dependence of the rate constant on the refractive index square is universal. Have the authors considered all possible radiative and non-radiative decay channels? This is important, since LLPS condensates are likely to be characterised by very high refractive indexes and it is vital to understand why the very well understood physics of the Strickler Berg formula does not work on the present probes (or, if it does, what other factors are superimposing

We would like to thank the reviewer for pointing this out. We refer the reviewer to the above answer, as it provides a thorough answer which covers answers to this question. In short, the well-understood Strickler-Berg relationship is indeed universal and it discusses the dependence of the radiative deexcitation channel of the fluorescence lifetime on the local refractive index, in the fluorophore microenvironment. This is important since the fluorophore in FPs is somewhat protected from the bulk, and a question can be raised as to whether the refractive index in the microenvironment of the fluorophore within the FP changes or not at different conditions. We suggest that differences can arise between different fluorophores depending on how they are caged/hidden from

the bulk solution. Since these differences depend, in part, on the structure of the FP, and on its structural stability, one explanation, which should undoubtedly be further tested, could be related to the different structural stabilities of the FPs, and how well the fluorophore is shielded from the exterior. To emphasize this point we added to the text references regarding the thermodynamic stabilities of the tested FPs (see above response).

3) Have the authors tested the effect of varying concentration of mCherry? Higher concentration quenching and consequential reduction in lifetime are often observed, and increased crowding can be exacerbating some of these effects.

We would like to thank the reviewer for this suggestion. The mCherry concentration used in the *in vitro* tests was not varied when tested in bulk solution. This is because we wanted to calibrate the fluorescence lifetime dependence as a function of changing additive concentrations. We also tested the fluorescence lifetimes of mCherry in precipitates, not only in the bulk solution. By default, in precipitates, the concentration of mCherry is larger than in bulk solution. Indeed, the experiment shows that precipitates of mCherry exhibit lower fluorescence lifetimes than in bulk solution, while prepared in the same bulk solution. However, this result alone is insufficient to claim that the fluorescence lifetimes of mCherry decrease with an increase in mCherry concentrations across all mCherry concentrations. In the precipitates, and in the HP1-tagged heterochromatin PS condensates, the mCherry FPs are probably within interaction proximities, which can induce the elevated crowdedness effects we observe. These results have been part of the original manuscript.

Altogether, our findings show that extremely high concentrations of mCherry itself, which will bring them to interaction distances, will lead to an enhanced reduction in fluorescence lifetime. Also, we did try varying mCherry concentrations for the *in vitro* precipitates' analyses and saw that at very low concentrations there isn't enough mCherry to form precipitates. At a bulk FP concentration of 100 nM, precipitates start to form at a given FVO and above it.

Importantly, we made sure to add the word "in solution" in the revised text, to make it clear that the precipitates were not dried up (see response to reviewer #1).

Finally, in the revised text, we added results from fluorescence anisotropy microscopy, which are thoroughly explained in my response to the reviewer's further question, below, as well as in my responses to reviewers #1-2. In short,

we now show that in undifferentiated ESCs, the fluorescence anisotropy is low, while in early-differentiated ESCs, the fluorescence anisotropy increase, suggesting changes in densities which lead to changes in homoFRET-related proximities. This can indicate the increase in the local concentration of FPs in different states of PS condensates, at local concentrations that bring FPs to <10 nm distances. Nevertheless, we explain in the revised text that since not only homoFRET might occur but also fluorescence lifetime changes, it is difficult to decouple homoFRET solely. Again, see my thorough response below.

Furthermore, the concentration of the fused protein in the nucleus would be poorly controlled, potentially causing heterogeneity observed in FLIM, which was interpreted as a variety of crowding environments for mCherry. At the same time, the same concentration variation is not likely to affect the bleached FCS data, taken as proof of the diffusivity of the constructs.

We thank the reviewer for this discussion. Obviously, the FP concentration scale spans from none to the densest packings, but at many FP concentrations, there is still no direct interaction of the FPs in solution. At the highest concentrations/densities that exist inside phase-separated foci, the FPs are within interaction proximities. Testing whether a correlation exists between pixel-wise fluorescence intensity and lifetime values, we find a weak correlation at most (new supplementary figure 14). It may well be that what we observe include fractions of FP-tagged proteins that are within interaction distances leading to the observed lifetime decreases, and other fractions that are concentrated but not to the level of interaction proximities. Regardless, if the heterogeneity observed in FLIM for undifferentiated ESCs was due to the increased concentration alone, we would expect to observe the same effect also in early-differentiating ESCs. However, in fact, fluorescence intensities within HP1alpha condensates in early-differentiated ESCs are higher than in undifferentiated ESCs, while the internal densities and nanocompaction levels are lower in early-differentiated ESCs relative to in undifferentiated ESCs, from fluorescence lifetime images, and now also from fluorescence anisotropy images.

Regarding the fluorescence lifetime data and the data from FCS, the two effects (diffusivity and bleaching and the heterogeneity in concentrations of mCherry-linked HP1 proteins) can be shown as partially dependent. While diffusivity is driven by temporal fluctuations in a given ROI, concentration changes can still be observed as spatial heterogeneity in our time-averaged images (time-averaged by the laser scanning parameters). This is yet another reason why we

test the crowding effects on FP lifetimes, first in a controlled manner *in vitro*. Still, it is clear that crowding in the bulk solution *in vitro* was induced solely by PEG, while in the nucleus there would be many factors that together lead to crowding of the FPs linked to HP1, some of them have the expected high densities in condensates, including the different concentrations of the mCherry themselves. However, from our *in vitro* estimation, it is crowding that mostly influences the fluorescence lifetime reductions in mCherry, whatever would be the molecular sources of crowding out of the myriad factors at play.

Regardless of this, while in FLIM we report heterogeneities spatially, in in-cell FCS we report diffusivities due to heterogeneities temporally per a given ROI, and the same is true for photobleaching curves.

We altered the text to discuss these insights (these points have been thoroughly described in the responses to reviewers #1-2).

4) I note that the change in lifetime is relatively minor for all but the highest crowding conditions (50% FVO) for both 'responsive' proteins, and unseen in cells. There is a further ambiguity with the fact that sometimes the lifetime changes upwards, eg in the presence of glycerol. This makes me think that other factors (polarity, dielectric?) might be at play.

The reviewer is correct in his/her observations. A few considerations are at play here:

- What is the minimal change in fluorescence lifetime that can be considered statistically significant, beyond the fitting precisions themselves
- Should only monotonic trends of fluorescence lifetime modulations be taken into account or also non-monotonic changes?

Both points raised here are interconnected. If titration of a given additive gives rise to non-monotonic changes in fluorescence lifetimes, and if these non-monotonic changes are fluctuating around a lifetime value that is close to the known literature value for the given FP, the breadth of these non-monotonic changes provides the means to define a fluorescence lifetime boundary, for which titration-dependent trends that are monotonic become statistically significant. These technicalities have been explained in the figures and in the text, and we have now extended them and added more clarifications. Text has been added to the results section "Fluorescence lifetime reductions in FPs due to elevated crowdedness levels" (page 5 in the revised manuscript):

Using these non-monotonic trends of mCherry lifetimes, we set a lower boundary of 1.44 ns (Fig. 1, dashed red line) as the lowest value for the fluorescence lifetime of mCherry.

Beyond this technical explanation, previous FLIM works have been reporting much smaller lifetime changes than the ones we report, which were accepted as true fluorescence lifetime changes, sometimes based solely on fitting precisions. In this work, we make the effort to calibrate a conservative boundary of what can be considered a fluorescence lifetime decrease that is statistically significant, relying on the expected monotonic decrease in fluorescence lifetimes, if indeed lifetimes decrease, and not only on fitting precisions. Beyond that, the observed fluorescence lifetime variations go beyond these boundaries enough to explain the observed lifetime heterogeneities. This is why in all figures, we add this boundary to distinguish different types of lifetime variations.

Finally, since our estimation of the fluorescence lifetime reduction effects performed in bulk solution was limited to 50% PEG, we tested higher PEG concentrations in PEG-driven precipitates which should have concentrations higher than in bulk, by design. Indeed, in these estimations, we observe even lower lifetimes than their values at the same %FVO PEG in bulk solution.

In conclusion, the observations reported here are interesting, and complemented by an attempted study of interesting biology that involves LLPS of chromatin. However, I feel that the characterisation of the probe is not complete and hence it is not possible to confidently assign the observed features to the crowding phenomena.

We hope that the explanations we provided for the suggestions/questions that were raised, as well as the additional references we have added to the introduction, the experimental data we added, and the results of our reanalyses (see also responses to reviewers #1 & #2), convincingly demonstrate the validity of our approach.

REVIEWERS' COMMENTS

Reviewer #1 (Remarks to the Author):

The authors have provided a truly improved version of the manuscript. In particular, I appreciate that they have now included a study using a previously published setting (Novo et al.) with MSR transcript depletion, which should shift the heterogeneous heterochromatin condensates to a more homogeneous form. This was indeed observed by the Lerner group. This result further strengthens their study and implementation of fluorescence lifetime imaging as a valuable tool to determine the nanoenvironment of molecules.

I have no further comments.

Reviewer #2 (Remarks to the Author):

The author has addressed the comments adequately and I therefore recommend publication.

Reviewer #3 (Remarks to the Author):

The authors thoroughly addressed the reviewers' comments and modified the text accordingly. I particularly enjoyed seeing additional data included in Fig 5-7 main text on HP1 α condensates that strengthen the methodology and conclusions. I have no further comments and recommend publication